# Enriched dietary saturated fatty acids induce trained immunity via ceramide production that enhances severity of endotoxemia and clearance of infection

Amy L Seufert[1], James W Hickman[1], Ste K Traxler[1], Rachael M Peterson[1], Trent A Waugh[1], Sydney J Lashley[2], Natalia Shulzhenko[3], Ruth J Napier[2,4], Brooke A Napier[1]*

[1]Department of Biology and Center for Life in Extreme Environments, Portland State University, Portland, United States; [2]VA Portland Health Care System, Portland, United States; [3]Department of Biomedical Sciences, Oregon State University, Corvallis, United States; [4]Department of Molecular Microbiology and Immunology, Oregon Health & Science University, Portland, United States

**Abstract** Trained immunity is an innate immune memory response that is induced by a primary inflammatory stimulus that sensitizes monocytes and macrophages to a secondary pathogenic challenge, reprogramming the host response to infection and inflammatory disease. Dietary fatty acids can act as inflammatory stimuli, but it is unknown if they can act as the primary stimuli to induce trained immunity. Here we find mice fed a diet enriched exclusively in saturated fatty acids (ketogenic diet; KD) confer a hyper-inflammatory response to systemic lipopolysaccharide (LPS) and increased mortality, independent of diet-induced microbiome and hyperglycemia. We find KD alters the composition of the hematopoietic stem cell compartment and enhances the response of bone marrow macrophages, monocytes, and splenocytes to secondary LPS challenge. Lipidomics identified enhanced free palmitic acid (PA) and PA-associated lipids in KD-fed mice serum. We found pre-treatment with physiologically relevant concentrations of PA induces a hyper-inflammatory response to LPS in macrophages, and this was dependent on the synthesis of ceramide. In vivo, we found systemic PA confers enhanced inflammation and mortality in response to systemic LPS, and this phenotype was not reversible for up to 7 days post-PA-exposure. Conversely, we find PA exposure enhanced clearance of *Candida albicans* in *Rag1*[-/-] mice. Lastly, we show that oleic acid, which depletes intracellular ceramide, reverses PA-induced hyper-inflammation in macrophages and enhanced mortality in response to LPS. These implicate enriched dietary SFAs, and specifically PA, in the induction of long-lived innate immune memory and highlight the plasticity of this innate immune reprogramming by dietary constituents.

## Editor's evaluation

This fundamental paper in mice convincingly demonstrates that a Western-type diet and the more extreme ketogenic diet for 2 weeks enhance monocyte-driven immune responsiveness. This leads to a deadly hyper-inflammatory state in the mice in response to an endotoxin challenge in vivo and enhances the clearance of pathogens. The paper is of interest to immunologists, infectious disease specialists, and nutritionists.

*For correspondence: brnapier@pdx.edu

Competing interest: The authors declare that no competing interests exist.

## Introduction

Historically, immune memory has been defined as a trait limited to the adaptive immune system; however, it is now well established that innate immune cells have the capacity for metabolic, epigenetic, and functional reprogramming that leads to long-lasting increases in host resistance to infection (*Netea et al., 2020*; *Kleinnijenhuis et al., 2012*; *Quintin et al., 2012*; *Saeed et al., 2014*). Specifically, trained immunity is an adaptation of innate host defense in vertebrates and invertebrates that results from exposure to a primary inflammatory stimulus and leads to a faster and greater response to a secondary challenge. Unlike adaptive memory responses, trained immunity does not require genome rearrangements, B and T lymphocytes, and receptors that recognize specific antigens (*Netea et al., 2020*; *Kleinnijenhuis et al., 2012*; *Quintin et al., 2012*; *Saeed et al., 2014*). Furthermore, trained immunity has been documented in organisms that lack canonical adaptive immune responses, such as plants and invertebrates, suggesting this is a primitive immune memory system that is conserved throughout vertebrates and invertebrates (*Lanz-Mendoza and Contreras-Garduño, 2022*).

The Bacillus Calmette-Guérin (BCG) vaccine and yeast β-glucans are canonical inducers of trained immunity in humans and stimulate long-lasting metabolic and epigenetic reprogramming of myeloid-lineage cells resulting in hyper-responsiveness upon restimulation with heterologous or homologous inflammatory stimuli. This innate immune memory has been shown to be heritable (*Katzmarski et al., 2021*) and can last up to months in humans and mice (*Netea et al., 2016*) and, thus, likely evolved to provide non-specific protection from secondary infections. Most recently, it was described that countries with higher rates of BCG vaccine at birth had fewer coronavirus disease 2019 (COVID-19) cases (*Covián et al., 2020*) making this immunological phenomenon extremely relevant. Importantly, it is easily ascertained that inflammatory hyper-responsiveness could be deleterious in the context of diseases where more inflammation can lead to greater pathology (e.g. acute septic shock, auto-immune disorders, and allergies). Thus, trained immunity can be regarded as a double-edged sword – providing increased resistance to tissue-specific infection but exacerbating diseases exacerbated by systemic inflammation. Consequently, identifying novel inducers of trained immunity will provide clinically relevant insight into harnessing innate immune cells to attain long-term therapeutic benefits in a range of infections and inflammatory diseases.

Typically, the primary inflammatory stimulus that initiates trained immunity is danger- or pathogen-associated molecular patterns (PAMPs); however, recent publications have shown that β-glucan found in mushrooms, baker's and brewer's yeast, wheat and oats, and unknown components of bovine milk can induce trained innate immune memory in monocytes in vitro (*Meena et al., 2013*; *van Splunter et al., 2018*). Our data reported here contribute to the growing evidence supporting the multifaceted immunoregulatory role of certain dietary constituents.

Currently, Westernized nations are increasingly dependent on diets enriched in saturated fatty acids (SFAs; *Swinburn et al., 2011*; *Popkin, 2006*; *Christ and Latz, 2019*), which have been shown to mimic PAMP effects on inflammatory cells, regulate innate immune cell function, and alter outcomes of inflammatory disease and infection (*Lancaster et al., 2018*; *Lumeng et al., 2007*; *Meikle and Summers, 2017*; *Reyes et al., 2021*). Specifically, we have shown the Western diet (WD), a diet enriched in sucrose and SFAs, correlates with increased disease severity and mortality in response to systemic LPS, independent of the diet-dependent microbiota, demonstrating the possibility that the dietary components of this diet may be driving the hyper-responsiveness to LPS (*Napier et al., 2019*). Currently, it is unknown if enriched dietary SFAs alone mediate trained immunity.

Our work presented herein identifies a ketogenic diet (KD) enriched exclusively in SFAs, and not sucrose, confers an increased systemic response to LPS independent of diet-associated microbiome, ketosis, or glycolytic regulation during disease, and alters inflammatory capacity and composition of the hematopoietic compartment. While others have shown that the WD induces trained immunity in atherosclerotic mice (Ldrl$^{-/-}$), we are the first to show that trained immunity, including its hallmark long-term persistence, can be induced in wild-type (WT) mice with exposure to enriched SFAs alone (*Christ et al., 2018*). A lipidomic analysis of blood fat composition after KD exposure revealed a significant increase of free palmitic acid (PA; C16:0) and fatty acid complexes containing PA. PA is known to act synergistically with LPS to enhance intracellular ceramide levels and proinflammatory cytokine expression in macrophages; however, it is currently unknown if ceramide, a bioactive sphingolipid (SG), specifically mediates a heightened inflammatory response to LPS following pre-exposure to PA (*Schilling et al., 2013*; *Zhang et al., 2017*). Here we find macrophages pre-treated with physiologically

relevant concentrations of PA followed by a secondary exposure to LPS lead to enhanced proinflammatory cytokine expression and release, which were reversible with the inhibition of ceramide.

We find that both short- and long-term exposure to PA, the predominant SFA found in high-fat diets, enhance systemic response to microbial ligands in mice even after a 7-day rest period from PA exposure. Thus, our data suggest exposure to PA leads to a long-lasting innate immune memory response in vivo (*Netea et al., 2016*). Importantly, trained immunity is induced when a primary inflammatory stimulus changes transcription of inflammatory genes, the immune status returns to basal levels, and challenge with a secondary stimulus enhances transcription of inflammatory cytokines at much higher levels than those observed during the primary challenge (*Divangahi et al., 2021*). While the dynamics of an initial inflammatory event induced by PA in vivo are not defined in this paper, we show that basal levels of *Tnf*, *Il6*, *Il1b,* and *Il10* in the blood of mice pre-exposed to PA were comparable to control mice immediately prior to endotoxin challenge, indicating that mice were not in a primed state prior to disease. This suggests that the hyper-inflammation and poor disease outcome we show in PA-exposed mice are not due to priming but a trained immune response.

The dual nature of trained immunity is also a hallmark feature of the phenomenon, in that non-specific innate immune memory can be either beneficial or detrimental depending on the disease context. The majority of research has demonstrated the protective role of trained immunity against a variety of infections, such as with BCG vaccination and B-glucan stimulation (*Quintin et al., 2012*; *Kaufmann et al., 2018*). Our work is unique because we focus on the detrimental role that trained immunity has on disease characterized by inflammatory dysregulation; however, we also highlight the beneficial nature of this novel phenotype by showing that when mice lacking adaptive immunity (*Rag1*$^{-/-}$) are pre-exposed to systemic PA, they exhibit enhanced clearance of kidney fungal burden compared to control mice.

We further identify a novel role of SFA-dependent intracellular ceramide required for the enhanced systemic response to microbial ligands, and show intervention with OA, a mono-unsaturated fatty acid that depletes PA-dependent ceramide, can reverse these phenotypes in macrophages and in vivo. Our data presented here highlight the dynamic plasticity of dietary intervention on inflammatory disease outcomes. These data are consistent with the current knowledge that SFAs and ceramide are immunomodulatory molecules, and build on these by highlighting a previously unidentified role of PA in driving long-lived trained immunity.

## Results

### Diets enriched in SFAs increase endotoxemia severity and mortality

To examine the immune effects of chronic exposure to diets enriched in SFAs on lipopolysaccharide (LPS)-induced endotoxemia, we fed age matched (6–8 weeks), female BALB/c mice either with a WD (enriched in SFAs and sucrose), a KD (enriched in SFAs and low-carbohydrate), or standard chow (SC; low in SFAs and sucrose), for 2 weeks (*Supplementary file 1*). We defined 2 weeks of feeding as chronic exposure, because this is correlated with WD- or KD-dependent microbiome changes and confers metaflammation in WD mice (*Napier et al., 2019*), sustained altered blood glucose levels in WD mice (*Figure 1—figure supplement 1A*), and elevated levels of ketones in the urine and blood in KD mice (*Figure 1—figure supplement 1B-C*). We then induced endotoxemia by a single intraperitoneal (i.p.) injection of LPS. We measured hypothermia as a measure of disease severity and survival to determine outcome (*Napier et al., 2019*; *Napier et al., 2016*; *Saito et al., 2003*). WD- and KD-fed mice showed significant and prolonged hypothermia, starting at 10 hr post-injection (p.i.), compared to the SC-fed mice (*Figure 1A*). In accordance with these findings, WD- and KD-fed mice displayed 100% mortality by 26 hr p.i. compared to 100% survival of SC-fed mice (*Figure 1B*). Hypoglycemia is a known driver of endotoxemia, and each of these diets has varying levels of sugars and carbohydrates (*Supplementary file 1*; *Raetzsch et al., 2009*; *Filkins and Cornell, 1974*). However, mice in all diet groups displayed similar levels of LPS-induced hypoglycemia during disease (*Figure 1—figure supplement 1D*), indicating that potential effects of diet on blood glucose were not a driver of enhanced disease severity.

Considering mice fed KD experience a shift toward nutritional ketosis, we wanted to understand if our phenotype was dependent on nutritional ketosis. 1,3-butanediol (BD) is a compound that induces ketosis by enhancing levels of the ketone β-hydroxybutyrate in the blood (*Goldberg et al., 2019*).

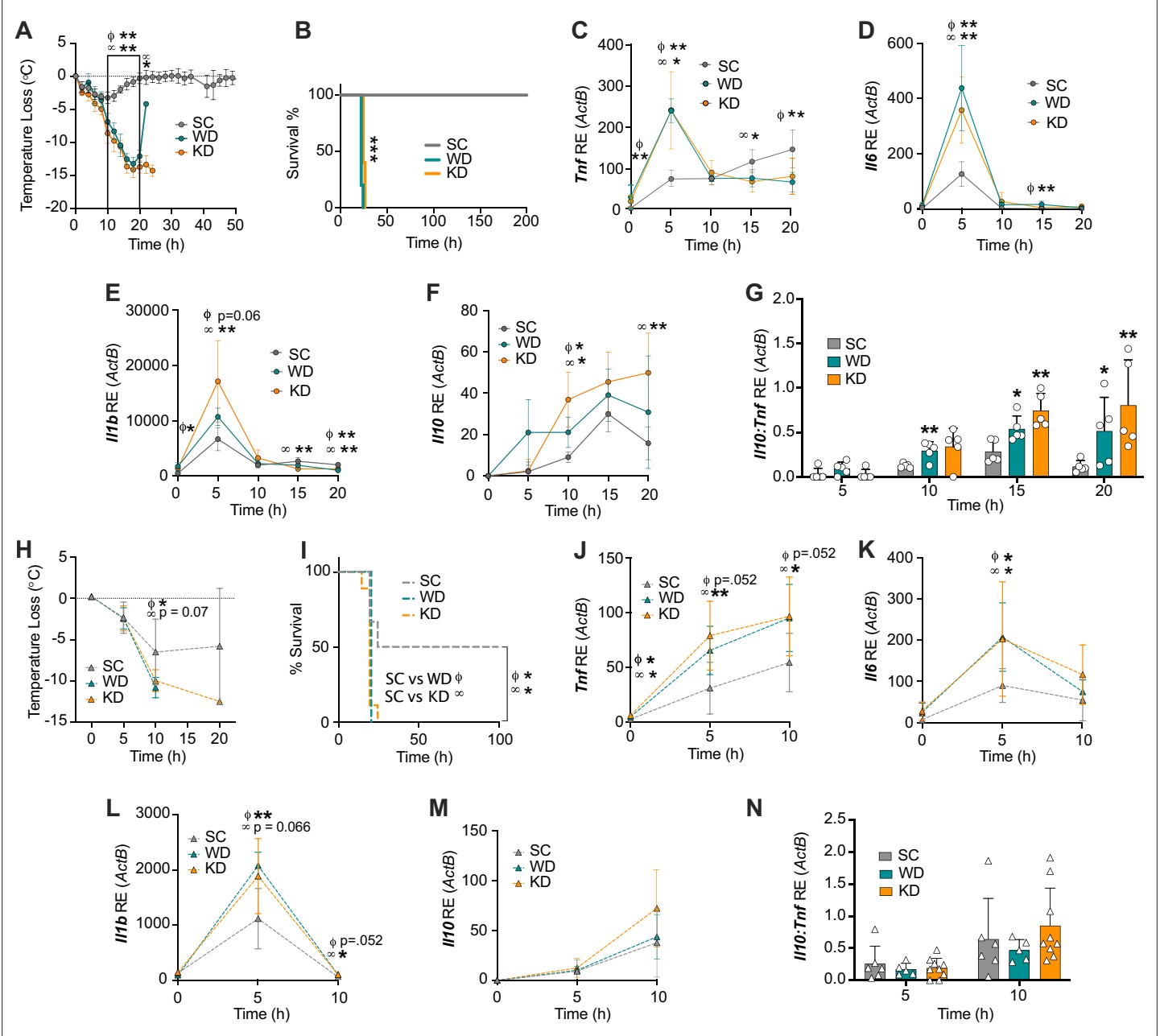

**Figure 1.** Diets enriched in saturated fatty acids lead to enhanced endotoxemia severity and altered systemic inflammatory profiles, independent of diet-associated microbiome. (A–G) Age-matched (6–8 weeks) female BALB/c mice were fed standard chow (SC), Western diet (WD), or ketogenic diet (KD) for 2 weeks and injected intraperitoneal (i.p.) with 6 mg/kg of lipopolysaccharide (LPS). (A) Temperature loss and (B) survival were monitored every 2 hr. At indicated times, 10–20 µL of blood was drawn via the tail vein, RNA was collected, and samples were assessed for expression of (C) *Tnf*, (D) *Il6*, (E) *Il1b*, and (F) *Il10* via qRT-PCR. (G) *Il10:Tnf* ratio was calculated for 5, 10, 15, and 20 hr post-injection (p.i.) with LPS. (H–N) Next, 19–23-week-old female and 14–23-week-old male and female germ-free C57BL/6 mice were fed SC, WD, or KD for 2 weeks and injected i.p. with 50 mg/kg of LPS. (H) Temperature loss and (I) survival were monitored every 5 hr p.i. (J–N) At indicated times, 10–20 µL of blood was drawn via the tail vein, RNA was collected, and samples were assessed for expression of (J) *Tnf*, (K) *Il6*, (L) *Il1b*, and (M) *Il10* via qRT-PCR. (N) *Il10:Tnf* ratio was calculated for 5 and 10 hr p.i. with LPS. For (A–G), all experiments were run three times, and data are representative of one experiment, n=5 per diet group. For (H–N) SC, n=6; WD, n=5; and KD, n=9; and data are representative of one experiment. For (A, C–G, H, and J–N) a Mann Whitney test was used for pairwise comparisons. For (B) and (I) a log-rank Mantel-Cox test was used for survival curve comparison. For all panels, *p<0.05; **p<0.01; ***p<0.001. For (C–E), Φ symbols indicate WD significance, and ∞ symbols indicate KD significance. Error bars shown mean ± SD.

The online version of this article includes the following source data and figure supplement(s) for figure 1:

**Source data 1.** Data and statistics for graphs depicted in *Figure 1A–N*.

*Figure 1 continued on next page*

*Figure 1 continued*

**Figure supplement 1.** Increase in disease severity in ketogenic diet (KD) mice is independent of ketosis.

**Figure supplement 1—source data 1.** Data for graphs depicted in *Figure 1—figure supplement 1A-H*.

Age matched (6–8 weeks), female BALB/c mice were fed for 2 weeks with KD, SC supplemented with saccharine and 1,3-butanediol (SC + BD), or SC-fed with the saccharine vehicle solution (SC + Veh). BD supplementation was sufficient to increase blood ketones (*Figure 1—figure supplement 1C*). We next injected LPS i.p. and found KD-fed mice showed significantly greater hypothermia, and increased mortality, compared to SC + BD and SC + Veh (*Figure 1—figure supplement 1E, F*). Though short-lived, when compared to SC + Veh, the SC + BD mice did confer an increase in hypothermia, suggesting that nutritional ketosis may play a minor role in KD-dependent susceptibility to endotoxemia (*Figure 1—figure supplement 1E, F*). Together these data suggest that diets enriched in SFAs promote enhanced acute endotoxemia severity, and this is independent of diet-dependent hypoglycemic shock or nutritional ketosis.

## Diets enriched in SFAs induce a hyper-inflammatory response to LPS and increased immunoparalysis

Endotoxemia mortality results exclusively from a systemic inflammatory response characterized by an acute increase in circulating inflammatory cytokine levels (e.g. TNF, IL-6, and IL-1β) from splenocytes and myeloid-derived innate immune cells (*Lewis et al., 2016*; *Wang et al., 2016*; *Radzyukevich et al., 2021*; *Zhang et al., 2012*). Additionally, pre-treatment of myeloid-derived cells with dietary SFAs has been shown to enhance inflammatory pathways in response to microbial ligands (*Schwartz et al., 2010*; *Fang et al., 2022*). Considering this, we hypothesized that exposure to enriched systemic dietary SFAs in WD- and KD-fed mice would enhance the inflammatory response to systemic LPS during the acute inflammatory response. 5 hr p.i., age matched (6–8 weeks), female BALB/c mice fed all diets showed induction of *Tnf, Il6,* and *Il1b* expression in the blood (*Figure 1C–E*). However, at 5 hr p.i., WD- and KD-fed mice experienced significantly higher expression of *Tnf* and *Il6* in the blood, compared with SC-fed mice, and WD-fed mice also showed significantly higher *Il1b* expression (*Figure 1C–E*), indicating that diets enriched in SFAs are associated with a hyper-inflammatory response to LPS.

Importantly, septic patients often present with two immune phases: an initial amplification of inflammation, followed by or concurrent with an induction of immune suppression (immunoparalysis), that can be measured by a systemic increase in the anti-inflammatory cytokine IL-10 (*Cheng et al., 2016*; *Nedeva et al., 2019*). Furthermore, in septic patients, a high IL-10:TNF ratio equates with the clinical immunoparalytic phase and correlates with poorer sepsis outcomes (*Gogos et al., 2000*; *van Dissel et al., 1998*). Interestingly, we found there was significantly increased *Il10* expression in WD- and KD-fed mice, compared to SC-fed mice (*Figure 1F*), and WD- and KD-fed mice had significantly higher *Il10:Tnf* ratios at 10–20 hr and 15–20 hr, respectively, compared to SC-fed mice (*Figure 1G*). These data conclude that mice exposed to diets enriched in SFAs show an initial hyper-inflammatory response to LPS, followed by an increased immunoparalytic phenotype, which correlates with enhanced disease severity, similar to what is seen in the clinic.

## Diets enriched in SFAs drive enhanced responses to systemic LPS independent of diet-associated microbiome

We have previously shown that WD-fed mice experience increased endotoxemia severity and mortality, independent of diet-associated microbiome (*Napier et al., 2019*). In order to confirm the increases in disease severity that correlated with KD were also independent of KD-associated microbiome changes, we used a germ-free (GF) mouse model. 19–23-week-old female and 14–23-week-old male and female GF C57BL/6 mice were fed SC, WD, and KD for 2 weeks followed by injection with 50 mg/kg of LPS, our previously established $LD_{50}$ in GF C57BL/6 mice (*Napier et al., 2019*). As we saw in the conventional mice, at 10 hr p.i. WD- and KD-fed GF mice showed enhanced hypothermia and mortality, compared to SC-fed GF mice (*Figure 1H, I*). These data show that, similar to WD-fed mice, the KD-associated increase in endotoxemia severity and mortality is independent of diet-associated microbiome.

Our previous studies (*Figure 1A–G*) in conventional mice were carried out in 6–8-week female mice on a BALB/c background. Importantly, genetic background and age differences can have large effects on LPS treatment outcome. The GF mice used in this study (*Figure 1H–N*) were on a C57BL/6 background, between the ages of 14 and 23 weeks. Thus, we confirmed WD- and KD-fed conventional C57BL/6 mice aged 20–21 weeks old show enhanced disease severity and mortality in an LPS-induced endotoxemia model (4.5 mg/kg), compared to mice fed SC, similar to what is seen in younger BALB/c mice (*Figure 1—figure supplement 1G, H*).

Additionally, to confirm that the hyper-inflammatory response to systemic LPS was independent of the WD- and KD-dependent microbiome, we measured systemic inflammation during endotoxemia via the expression of *Tnf*, *Il6*, and *Il1b* in the blood at 0–10 hr p.i. We found, WD- and KD-fed GF mice displayed enhanced expression of *Tnf* and *Il1b* at 5–10 hr, and significantly enhanced expression of *il-6* at 5 hr, compared to SC-fed GF mice (*Figure 1J–L*). Interestingly, *Il10* expression and the *Il10:Tnf* ratio were not significantly different throughout all diets, suggesting the SFA-dependent enhanced immunoparalytic phenotype is dependent on the diet-associated microbiomes in WD- and KD-fed mice (*Figure 1M–N*). These data demonstrate that the early hyper-inflammatory response, but not the late immunoparalytic response, to LPS associated with enriched dietary SFAs is independent of the diet-dependent microbiota.

## A diet enriched exclusively in SFAs induces trained immunity

Thus far we find feeding diets enriched in SFAs (WD and KD) leads to enhanced expression of inflammatory cytokines in the blood after treatment with systemic LPS, suggesting that the SFAs may be inducing an innate immune memory response that leads to a hyper-inflammatory response to secondary challenge. Specifically, trained immunity is an innate immune memory response characterized by reprogramming of myeloid cells by a primary inflammatory stimulus, that then responds more robustly to secondary inflammatory challenge. Trained immunity has been shown to mediate cell subtypes within the HSC compartment that gives rise to "trained" myeloid progeny for weeks to years (*De Zuani and Frič, 2022*). A previous study in *Ldlr*$^{-/-}$ mice has shown 4 weeks of WD feeding significantly enhances multipotent progenitors (MPPs) and granulocyte and monocyte precursors (GMPs) and skews development of GMPs toward a monocyte lineage that is primed to respond with a hyper-inflammatory response to LPS (*Christ et al., 2018*). Currently, it is unknown if diets enriched in SFAs fed to WT mice can induce changes within the HSC compartment or long-lasting trained immunity.

In order to determine the impact of dietary SFAs on bone marrow reprogramming in vivo, we next evaluated HSCs and progenitor cells via flow cytometry from age-matched (6–8 weeks) female WT BALB/c mice fed SC, WD, and KD for 2 weeks. Using previously published panels for analyzing HSC populations in the bone marrow (*Kaufmann et al., 2018*; *Vazquez et al., 2015*; *Nowlan et al., 2020*), we collected bone marrow and measured relative proportions of long-term HSCs (LT-HSCs; CD201$^+$CD27$^+$CD150$^+$CD48$^-$), short-term HSCs (ST-HSCs; CD201$^+$CD27$^+$CD150$^+$CD48$^+$), and MPPs (CD201$^+$CD27$^+$CD150$^-$CD48$^+$) (*Figure 2A and B*). Strikingly, we find that KD-fed mice showed significantly enhanced ST- and LT-HSCs, and MPPs compared to SC-fed mice (*Figure 2C*). Unlike previously reported in *Ldlr*$^{-/-}$ mice, there was no significant change in ST-HSCs, LT-HSCs, or MPPs within WD-fed WT mice (*Figure 2C*). Furthermore, we did not see a significant increase in MPP3s for WD-fed mice, as previously published for Ldlr$^{-/-}$ mice (*Christ et al., 2018*), or KD-fed mice; however, this may be due to the difference in genetic backgrounds, or length of diet administration (*Figure 2—figure supplement 1A*). These data are the first to show that the KD, a diet solely enriched in SFAs, alters hematopoiesis by enhancing expansion and differentiation of HSCs, similar to previously described inducers of trained immunity.

Furthermore, it is unknown if enriched dietary SFAs lead to long-lasting functional reprogramming associated with trained immunity that leads to a hyper-inflammatory response. Thus, we fed age-matched (6–8 weeks) female BALB/c mice SC, WD, or KD for 2 weeks, isolated bone marrow, differentiated into BMDMs for 7 days, and analyzed baseline inflammation and response to LPS. We found that untreated BMDMs isolated from mice fed SC and WD showed no significant differences in TNF or IL-6, and those from KD-fed mice showed a modest increase only in IL-6 compared to BMDMs from SC-fed mice (*Figure 2—figure supplement 1B*). However, when BMDMs were stimulated with LPS for 24 hr ex vivo, BMDMs from WD- and KD-fed mice showed significantly higher secretion of TNF, and only those from KD-fed mice showed significantly enhanced IL-6 secretion (*Figure 2D–E*). These data

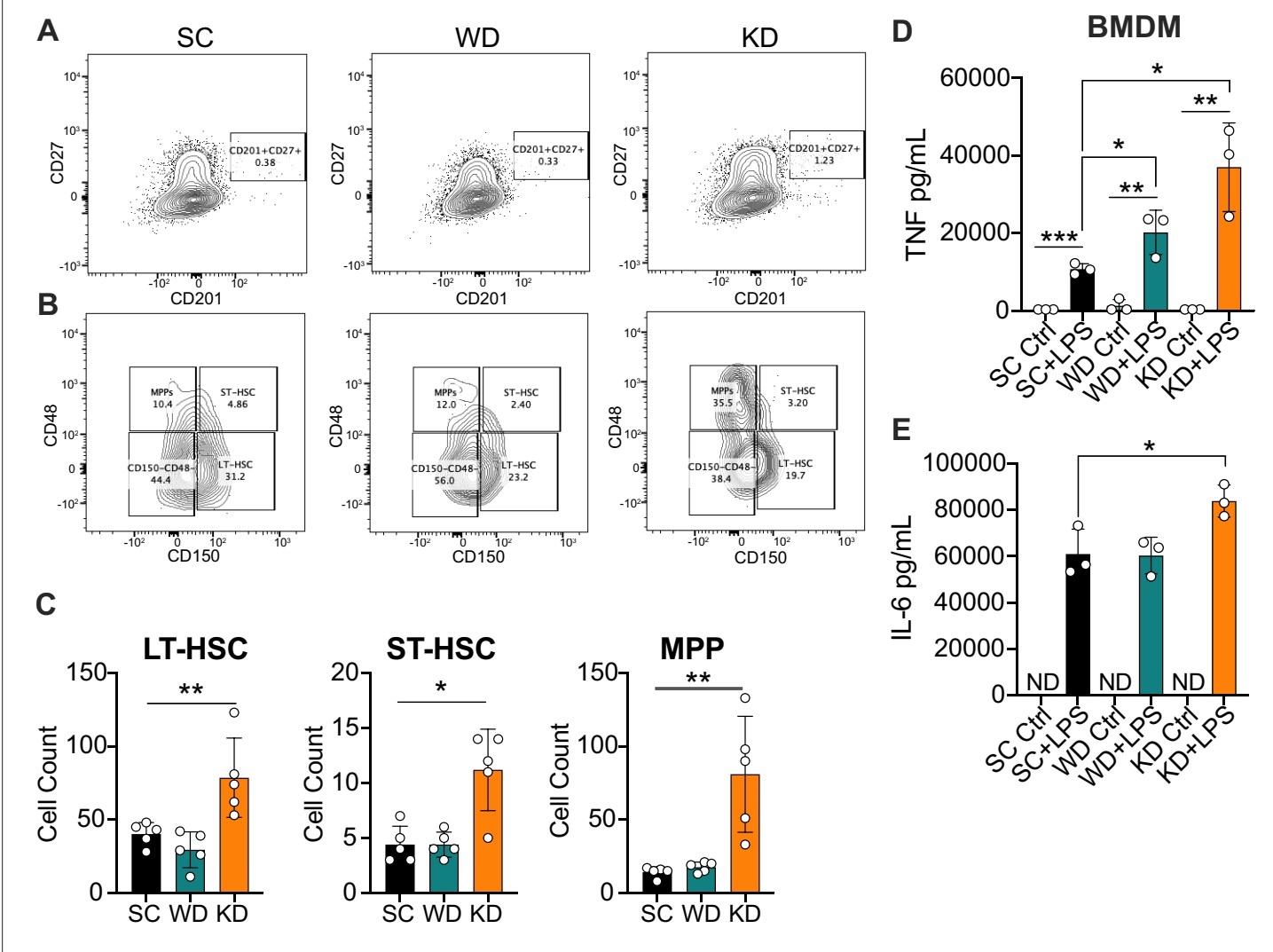

**Figure 2.** Ketogenic diet (KD) feeding alters HSC populations and bone marrow-derived macrophages (BMDMs) from KD-fed mice show a hyper-inflammatory response to lipopolysaccharide (LPS) ex vivo. Bone marrow was extracted from the femurs and tibias of age-matched (6–8 weeks) female BALB/c mice fed standard chow (SC), Western diet (WD), or KD for 2 weeks. (**A**) Fluorescence-Activated Cell Sorting (FACS) plots of total HSCs (CD201⁺CD27⁺) and (**B**) LT-HSCs, ST-HSCs, and multipotent progenitors (MPPs) from mice fed SC, WD, or KD for 2 weeks. Quantification of (**C**) the total numbers of LT- and ST-HSCs, and MPPs in bone marrow from mice fed SC, WD, or KD for 2 weeks. Next, BMDMs were plated at 5×10^6 cells/mL and differentiated for 7 days in media supplemented with macrophage colony-stimulating factor. Cells were split and plated in 24-well plates to adhere for 12 hr and treated with media (Ctrl) or LPS (24 hr; 10 ng/mL). Supernatants were assessed via ELISA for (**D**) TNF and (**E**) IL-6 secretion at 24 hr post-LPS treatment. IL-6 Ctrl supernatants were below the limit of detection; ND = no data. For (**A-E**), all experiments were run three times, and data are representative of one experiment, n=5 per diet group. (**C**) A Mann Whitney test was used for pairwise comparisons. (**D, E**) For all plates, all treatments were performed in triplicate, and a student's t-test was used for statistical significance. *, p<0.05; **, p<0.01; ***, p<0.001; ****, p<0.0001. Error bars show the mean ± SD.

The online version of this article includes the following source data and figure supplement(s) for figure 2:

**Source data 1.** Data and statistics for graphs depicted in *Figure 2A–E*.

**Figure supplement 1.** Ketogenic diet (KD) does not alter multipotent progenitor (MPP) differentiation or basal inflammation in bone marrow-derived macrophages (BMDMs), and monocytes and splenocytes show a hyper-inflammatory response to lipopolysaccharide (LPS) ex vivo.

**Figure supplement 1—source data 1.** Data for graphs depicted in *Figure 2—figure supplement 1*.

**Figure supplement 2.** Gating strategy for HSCs, related to *Figure 2*.

show that diets enriched in SFAs are inducing long-lasting inflammatory reprogramming of myeloid cells in vivo, and that reprogramming takes place within the bone marrow.

Importantly, monocytes and splenocytes are necessary for induction of systemic inflammatory cytokines during endotoxemia (*Radzyukevich et al., 2021*; *Zhang et al., 2012*). Thus, we wanted to assess if enriched dietary SFA induces in vivo reprogramming of monocytes and splenocytes, leading to an enhanced response to LPS ex vivo. First, we fed age-matched (6–8 weeks) female BALB/c mice SC, WD, or KD for 2 weeks, isolated bone marrow monocytes (BMMs) via magnetic negative selection using bone marrow extracted from femurs and tibias, and determined baseline expression of inflammatory cytokines. We found that prior to ex vivo LPS stimulation, BMMs isolated from mice fed SC, WD, or KD showed no significant difference in *Tnf* expression, and *Il6* expression was significantly decreased in BMMs from KD-fed mice (*Figure 2—figure supplement 1C*). However, when BMMs were stimulated with LPS for 2 hr ex vivo, those from KD-fed mice showed significantly higher expression of *Tnf* and *Il6*, while those from WD-fed mice exhibited no significance in expression compared to SC-fed mice (*Figure 2—figure supplement 1D*). Similarly, we isolated splenocytes from SC-, WD-, and KD-fed mice and found no difference between homeostatic inflammation of splenocytes between diets, but a significantly enhanced expression of *Tnf* in the splenocytes of KD-fed mice, and not WD-fed mice, challenged with LPS (2 hr) compared to splenocytes from SC-fed mice (*Figure 2—figure supplement 1E, F*).

These data show the KD stimulates expansion of HSC populations and skew differentiation of myeloid progenitors that then give rise to macrophages with enhanced inflammatory potency (*Figure 2A–E*; *Figure 2—figure supplement 2*). Furthermore, these data suggest that BMDMs, BMMs, and splenocytes from WD- and KD-fed mice are not more inflammatory at homeostasis; however, when challenged with LPS, KD feeding confers a hyper-inflammatory response. Together, our results suggest the KD, a diet that comprised 90.5% SFAs, leads to reprogramming of the HSC compartment and long-lasting trained immunity.

## PA and PA-associated fatty acids are enriched in the blood of KD-fed mice

It is known that the SFAs consumed in the diet determine the SFA profiles in the blood (*Dougherty et al., 1987*; *Skeaff et al., 2006*; *Zöllner and Tatò, 1992*), and that these SFAs have the potential to be immunomodulatory. Thus, we next wanted to identify target SFAs enriched in the blood of mice fed a diet exclusively enriched in SFAs that may be altering the systemic inflammatory response to LPS. Considering that the KD is enriched in SFAs and not sucrose, and that KD-fed mice showed distinct HSC alterations and LPS-induced hyper-inflammation in BMDMs, BMMs, and splenocytes treated ex vivo, the subsequent studies were performed exclusively on KD-fed mice. We used mass spectrometry lipidomics to create diet-dependent profiles of circulating fatty acids in SC- and KD-fed mice (*Choi et al., 2018*). Age matched (6–8 weeks), female BALB/c mice were fed SC or KD for 2 weeks, then serum samples were collected and analyzed using qualitative tandem liquid chromatography quadrupole time of flight mass spectrometry. We used principal component analysis (PCA) to visualize how samples within each data set clustered together according to diet, and how those clusters varied relative to one another in abundance levels of free fatty acids (FFA), triacylglycerols (TAG), and phosphatidylcholines (PC). For all three groups of FAs, individual mice grouped with members of the same diet represented by a 95% confidence ellipse with no overlap between SC- and KD-fed groups (*Figure 3A–C*). These data indicate that 2 weeks of KD feeding are sufficient to alter circulating FFAs, TAGs, and PCs, and that SC- and KD-fed mice display unique lipid blood profiles. Similarly, the relative abundance of SGs in SC- and KD-fed mice displayed unique diet-dependent profiles with no overlapping clusters, and abundance of specific SGs was significantly higher in the serum of KD-fed mice compared to SC-fed mice (*Figure 3—figure supplement 1A, B*). Though the independent role of each FFA, TAG, PC, and SG species has not been clinically defined, each are classes of lipids that when accumulated is associated with metabolic diseases, which have been shown to enhance susceptibility to sepsis and exacerbate inflammatory disease (*Meikle and Summers, 2017*; *I S Sobczak et al., 2019*; *Sokolowska and Blachnio-Zabielska, 2019*; *Papadimitriou-Olivgeris et al., 2016*).

Importantly, we identified a significant increase in multiple circulating FFAs within the KD-fed mice, compared to the SC-fed mice, many of which were SFAs (*Figure 3D*). Interestingly, in KD-fed mice we found a significant increase in free PA (C16:0), an immunomodulatory SFA that is found naturally

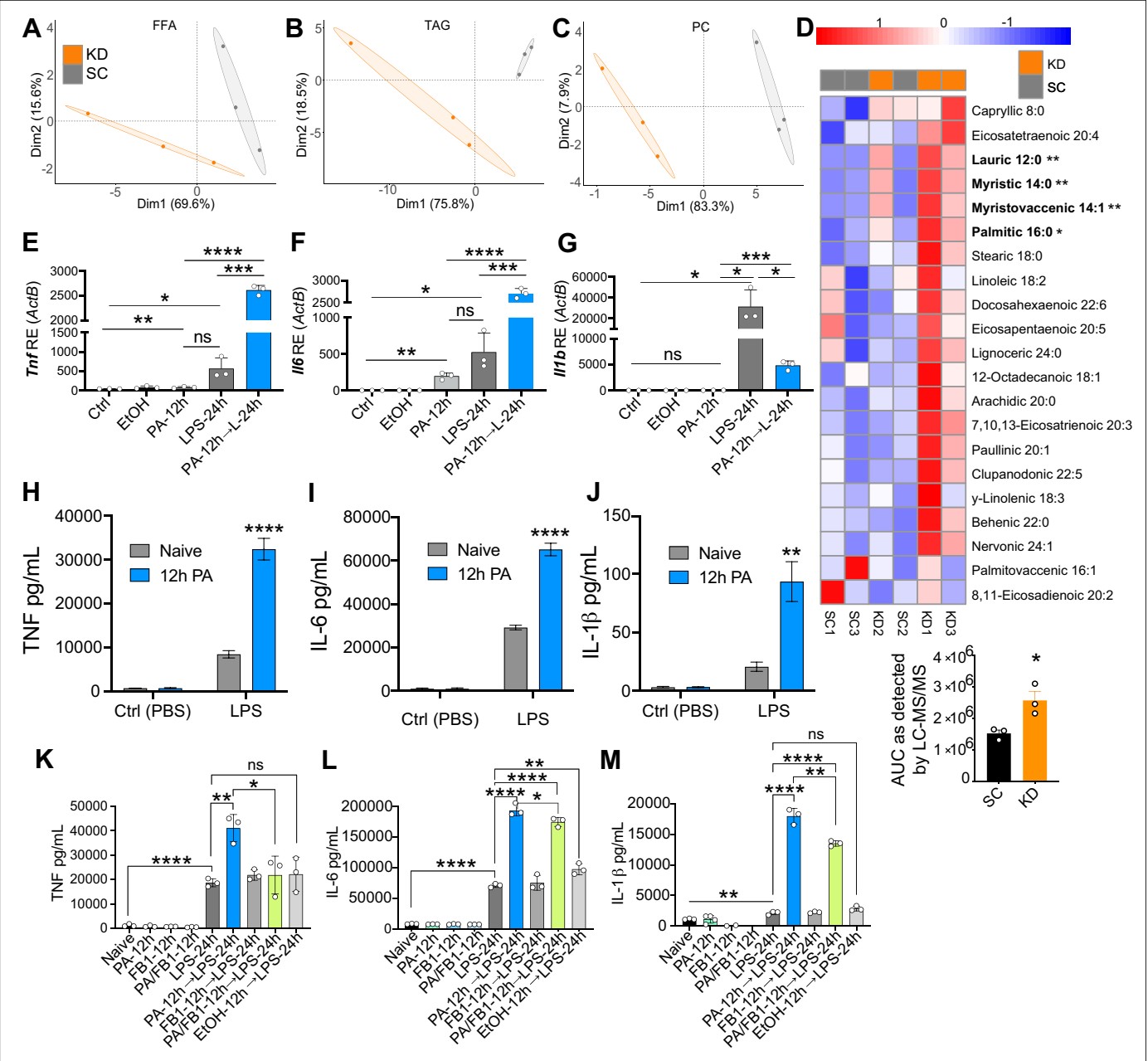

**Figure 3.** Ketogenic diet (KD) alters lipid profiles, and palmitic acid (PA) is mediating a hyper-inflammatory response to secondary challenge with lipopolysaccharide (LPS). Data points represent single animal samples, and colors represent groups fed standard chow (SC; gray) or KD (orange) diets for 2 weeks. A 95% confidence ellipse was constructed around the mean point of each group for (**A**) free fatty acids (FFA), (**B**) triglycerides (TAG), and (**C**) phosphatidylcholines (PC). (**D**) Heatmap analysis of FFA in SC and KD mice. Components that are significantly different between the two groups are in bold. Below the heatmap is a comparison of PA 16:0 peak area detected by Liquid Chromatography Quadruple Time of Flight Mass Spectrometry (LC-QToF MS/MS) between SC and KD groups; AUC = area under the curve. Statistical significance is determined by unpaired two-tailed t-test between SC and KD groups with n=3 per group. Primary bone marrow-derived macrophages (BMDMs) were isolated from age-matched (6–8 weeks) C57BL/6 female and male mice. BMDMs were plated at $1×10^6$ cells/mL and treated with either ethanol (ethanol (EtOH); media with 0.83% ethanol), media (Ctrl for LPS), or LPS (10 ng/mL) for 12 hr, or PA (PA stock diluted in 0.83% EtOH; 1 mM PA conjugated to 2% bovine serum albumin [BSA]) for 12 hr, with and without a secondary challenge with LPS. After indicated time points, RNA was isolated, and expression of (**E**) *Tnf*, (**F**) *Il6*, and (**G**) *Il1b* was measured via qRT-PCR. BMDMs were plated at $1×10^6$ cells/mL and treated with either ethanol (EtOH; media with 0.83% ethanol), media (Naïve), or 1 mM PA for 12 hr followed by PBS (control) or LPS (10 ng/mL). Supernatants were assessed via ELISA for (**H**) TNF, (**I**) IL-6, and (**J**) IL-1β secretion. Next, BMDMs were plated at $1×10^6$ cells/mL and treated with either media (Ctrl), LPS (10 ng/mL) for 24 hr, PA (PA stock diluted in 0.83% EtOH; 0.5 mM PA conjugated to 2% BSA) for 12 hr, Fumonisin B1 (FB1; 10 μM; diluted in 0.14% EtOH) or EtOH (0.97% to mimic simultaneous PA/FB1 treatment). Controls for all treatments are shown next to experimental groups treated additionally with LPS (10 ng/mL) for 24 hr. Supernatants were assessed via ELISA for (**K**) TNF, (**L**) IL-6, and (**M**)

*Figure 3 continued on next page*

Figure 3 continued

IL-1β secretion. For all plates, all treatments were performed in triplicate. For all panels, a student's t-test was used for statistical significance. *, p<0.05; **, p<0.01; ***, p<0.001; ****, p<0.0001. Error bars show the mean ± SD.

The online version of this article includes the following source data, source code, and figure supplement(s) for figure 3:

**Source code 1.**

**Source data 1.** Data and statistics for graphs depicted in *Figure 3A–M*.

**Figure supplement 1.** Principal component analysis and heatmap analysis of sphingolipid lipidomic data in mouse serum samples.

**Figure supplement 1—source data 1.**

**Figure supplement 1—source code 1.**

**Figure supplement 2.** Physiological levels of palmitic acid (PA) induce a hyper-inflammatory response to secondary challenge with lipopolysaccharide (LPS) in macrophages.

**Figure supplement 2—source data 1.** Data for graphs depicted in *Figure 3—figure supplement 2A-D*.

**Figure supplement 3.** Cytotoxicity as determined by lactate dehydrogenase (LDH) release from bone marrow-derived macrophages (BMDMs) pre-treated with palmitic acid (PA) followed by lipopolysaccharide (LPS) stimulation.

**Figure supplement 3—source data 1.** Data for graphs depicted in *Figure 3—figure supplement 3A, B*.

in animal fats, vegetable oils, and human breast milk (*Mancini et al., 2015*) and is eightfold enriched in KD (*Figure 3D*, *Supplementary file 1*). Additionally, PA-containing TAGs and PCs were significantly elevated in KD-fed mice serum, compared to SC-fed mice (*Figure 3—figure supplement 1C, D*). These data indicate that KD feeding not only enhances levels of freely circulating PA, but also enhances the frequency PA is incorporated into other lipid species in the blood.

## PA enhances macrophage inflammatory response to LPS

Many groups have shown that PA alone induces a modest, but highly reproducible increase in the expression and release of inflammatory cytokines in macrophages and monocytes (*Lancaster et al., 2018*; *Korbecki and Bajdak-Rusinek, 2019*). However, it remains unknown if PA can act as a primary inflammatory stimulus to induce a hyper-inflammatory response to a secondary heterologous stimulus in primary cells. Thus, we next wanted to determine if pre-exposure to physiologically relevant concentrations of PA altered the macrophage response to a secondary challenge with LPS. Current literature indicates a wide range of serum PA levels, between 0.3 and 4.1 mM, reflects a high-fat diet in humans (*Abdelmagid et al., 2015*; *Liu et al., 2015*; *Gallego et al., 2018*; *Buchanan et al., 2021*; *Perreault et al., 2014*). We aimed to use a physiologically relevant concentration of PA reflecting a human host for our in vitro studies, thus we treated primary bone marrow-derived macrophages (BMDMs) with and without 1 mM of PA containing 2% bovine serum albumin (BSA) for 12 hr, removed the media, subsequently treated with LPS (10 ng/mL) for an additional 24 hr, and measured expression and release of TNF, IL-6, and IL-1β. Importantly, the BSA dissolved in the media used for PA treatment solutions was endotoxin- and FA-free to ensure aberrant TLR signaling would not occur via BSA contamination, and fresh PA was conjugated to BSA-containing media immediately prior to use. We found that BMDMs pre-treated with PA (1 mM) for 12 hr expressed significantly higher levels of *Tnf* and *Il6* in response to secondary treatment with LPS, compared to naïve BMDMs (*Figure 3E and F*). *Il1b* expression was significantly lower in cells pre-treated with PA (*Figure 3G*); however, secretion of TNF, IL-6, and IL-1β was enhanced in BMDMs pre-treated with PA (1 mM) for 12 hr and challenged with LPS (*Figure 3H–J*). We found a similar enhanced *Il6* and *Tnf* expression in response to LPS in BMDMs treated with PA (1 mM) for twice the length of exposure (24 hr), and *Il-1b* expression was decreased (*Figure 3—figure supplement 2A-C*).

Furthermore, we pre-treated BMDMs with a concentration of PA that reflects the lower range of physiologically relevant serum levels and found 0.5 mM of PA induced significantly higher expression of *Tnf*, *Il6*, and *Il1b* after 12-hr challenge with LPS; however, only *Tnf* and *Il6* were significantly enhanced after 24-hr LPS treatment, compared to naive BMDMs treated with LPS (*Figure 3—figure supplement 2D-I*).

Importantly, PA treatment can induce apoptosis and pyroptosis in various cell types (*Borradaile et al., 2006*; *Li et al., 2018*; *Ly et al., 2017*; *Tao et al., 2021*); however, we found only an average of 3.4 and 4.4% of cell death after a 12-hr or 24-hr incubation, respectively, with PA (1 mM) and

subsequent 24 hr of LPS treatment or control media (*Figure 3—figure supplement 3A, B*). These data demonstrate PA pre-treatment of macrophages induces a hyper-inflammatory response to LPS independent of cell death, suggesting PA is sensitizing macrophages to secondary inflammatory challenge.

Thus, we conclude that both 12- and 24-hr pre-treatments with 0.5 mM or 1 mM of PA conjugated to 2% BSA are sufficient to induce reprogramming of macrophages and alter the response to stimulation with a heterologous ligand. Additionally, these data demonstrate that even serum concentrations of PA that are at the lower end of the spectrum for humans consuming a high-fat diet pose a risk for inflammatory dysfunction.

## Diverting ceramide synthesis inhibits the PA-dependent hyper-inflammatory response to LPS in macrophages

PA treatment of various cell types diverts cellular metabolism toward the synthesis of the toxic metabolic byproducts: diacylglycerols and ceramide (*Palomer et al., 2018*). PA-induced ceramide synthesis has specifically been demonstrated to enhance inflammation (*Schilling et al., 2013*; *Zhang et al., 2017*; *Schwartz et al., 2010*; *Jin et al., 2013*). Considering this, we wanted to determine the role of enhanced macrophage ceramide production in driving PA-induced hyper-inflammatory response to LPS. Thus, we treated BMDMs simultaneously with PA (0.5 mM) and a ceramide synthase inhibitor Fumonisin B1 (FB1; 10 µM), for 12 hr, removed the media, subsequently treated with LPS (10 ng/mL) for an additional 24 hr, and measured release of TNF, IL-6, and IL-1β. We found that BMDMs pre-treated simultaneously with PA and FB1 for 12 hr expressed significantly lower levels of TNF, IL-6, and IL-1β secretion in response to LPS, compared to BMDMs pre-treated with only PA (*Figure 3K–M*). We conclude that ceramide synthesis induced by PA is required for the macrophage hyper-inflammatory response to secondary challenge with LPS.

## PA is sufficient to increase endotoxemia severity and systemic hyper-inflammation

Considering the drastic effect of PA on macrophage response to secondary challenge with LPS, we next wanted to understand if exposure to PA alone is sufficient to induce a hyper-inflammatory response during endotoxemia in vivo. We answered this question using age-matched (6–8 weeks) female BALB/c mice fed SC for 2 weeks, by mimicking systemic PA levels found in serum of humans on high-fat diet via a single i.p. injection of ethyl palmitate (750 mM), and then after 12 hr, challenging with LPS i.p. (*Eguchi et al., 2012*). Similar to previous publications, we find that a 750-mM i.p. injection of ethyl palmitate enhances free PA levels in the serum to 173–425 µM compared to Veh-treated mice with 110–250 µM (*Figure 4—figure supplement 1A*). Important to note, free PA is only transiently enhanced by systemic application and is quickly (<1 hr) taken up by peripheral tissues; thus, our detected free serum levels are most likely an underestimation of transient systemic PA (*Black, 2007*; *Mansbach and Gorelick, 2007*; *Karavolos et al., 2008*).

Interestingly, after LPS challenge, PA-treated mice experienced increased disease severity as indicated by their significant decline in temperature compared to Veh mice (*Figure 4A*). Similar to WD- and KD-fed mice, PA-treated mice also exhibited enhanced mortality, compared to Veh mice (*Figure 4B*). Importantly, mice injected with PA for shorter time periods (0, 3, and 6 hr) and then challenged with LPS did not exhibit increased disease severity or poor survival outcome (*Figure 4—figure supplement 1B, C*), concluding that a 12-hr pre-treatment with PA is required for an increase in disease severity.

Next, we measured systemic inflammatory status during disease and found similar to KD-fed mice, the 12-hr PA-pre-treated mice showed significantly enhanced expression of *Tnf* (5 hr and 10 hr) and *Il6* (5 hr) post-LPS challenge, compared to Veh control (*Figure 4C and D*). Expression of *Il-1b* trended upward but was not significantly upregulated in 12-hr PA-pre-treated mice, compared to Veh-treated mice (*Figure 4E*). Importantly, as a control we looked at LPS-induced hypoglycemia in PA-treated mice, and 12-hr pre-treatment with PA did not alter LPS-induced hypoglycemia (*Figure 4—figure supplement 1D*), indicating that diet-dependent hypoglycemic shock was not a driver of endotoxemia severity in PA-treated mice. Thus, exposure to PA to mimic systemic levels found in humans eating high-fat diets is sufficient to drive enhanced inflammation and disease severity in mice stimulated with endotoxin, and this effect is dependent on length of PA exposure.

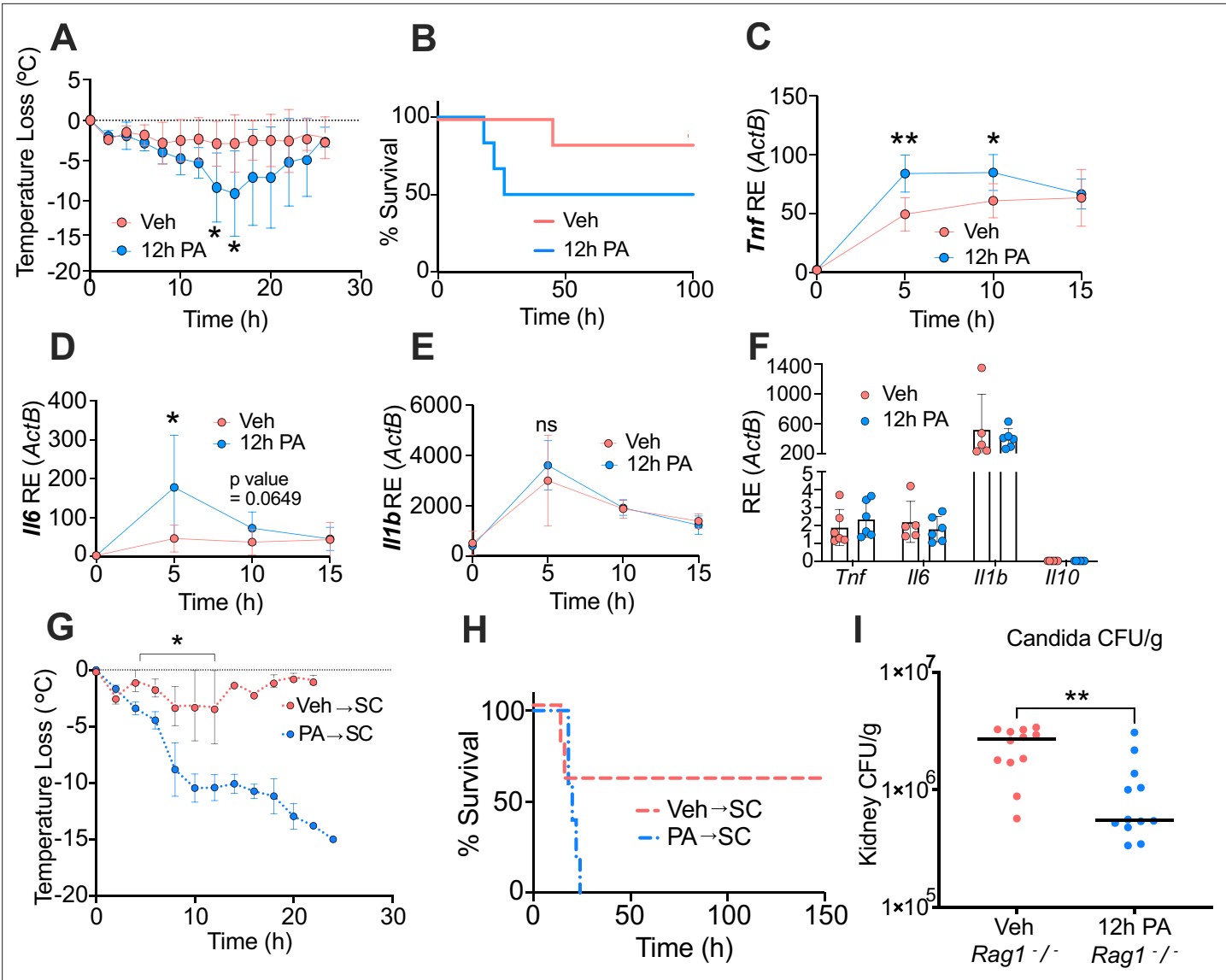

**Figure 4.** Palmitic acid (PA) acts as a novel mediator of trained immunity by inducing a hyper-inflammatory response lipopolysaccharide (LPS)-induced endotoxemia and enhancing clearance of *Candida albicans* infection. Age-matched (6–8 weeks) female BALB/c mice were fed standard chow (SC) for 2 weeks and injected intraperitoneal (i.p.) with ethyl palmitate (PA, 750 mM) or vehicle (Veh) solutions 12 hr before i.p. LPS injections (10 mg/kg). (**A**) Temperature loss was monitored every 2 hr as a measure of disease severity or (**B**) survival. At indicated times blood was collected via the tail vein, RNA was extracted, and samples were assessed for expression of (**C**) *Tnf*, (**D**) *Il6*, and (**E**) *Il1b* via qRT-PCR. (**F**) Blood was collected via the tail vein from Veh and PA pre-treated (12-hr PA) mice immediately prior to LPS injection, and samples were assessed for expression of *Tnf, Il6, Il1b*, and *Il10* via qRT-PCR. Additionally, age-matched (6–8 weeks) female BALB/c mice fed SC, injected i.p. with ethyl palmitate (PA, 750 mM) or Veh solutions every day for 9 days, and then rested for 7 days before i.p. LPS injections (10 mg/kg) (**G**) Temperature loss and (**H**) survival were monitored during endotoxemia. (**I**) Age-matched (8–9 weeks) female *Rag1⁻/⁻* mice were injected i.p. with ethyl palmitate (PA, 750 mM) or Veh solutions 12 hr before intravenous *C. albicans* infection. Fungal burden of kidneys from Veh and PA pre-treated (12-hr PA) mice 24 hr after *C. albicans* infection. For (**A–F**), experiments were run three times, and data are representative of one experiment, n=3 mice/group. For (**G, H**), experiments were run twice, and data are representative of one experiment, n=5 mice/group. For (**I**), experiments were run three times, and data are representative of one experiment, n=6 mice/group. For (**A**), (**C–E**), (**G**), and (**I**), a Mann Whitney test was used for pairwise comparisons. For (**B**) and (**H**), a log-rank Mantel-Cox test was used for survival curve comparison. For all panels, *, p<0.05; **, p<0.01; ***, p<0.001; ****, p<0.0001. Error bars shown mean ± SD.

The online version of this article includes the following source data and figure supplement(s) for figure 4:

**Source data 1.** Data and statistics for graphs depicted in *Figure 4A–I*.

**Figure supplement 1.** Palmitic acid (PA) intraperitoneal (i.p.) injections enhance serum PA concentrations, and PA-induced trained immunity is time-dependent.

**Figure supplement 1—source data 1.** Data for graphs depicted in *Figure 4—figure supplement 1A-D*.

## PA induces long-lived hyper-responsiveness to LPS and enhanced clearance of fungal infection

Our data show that pre-treatment with systemic PA alone enhances endotoxemia severity in vivo and inflammatory responses of macrophages to a secondary and heterologous stimulus in vitro. This form of regulation resembles trained immunity; however, it remains unclear if PA is inducing trained immunity in vivo. We first evaluated the basal level expression of *Tnf, Il6,* and *Il1b* in mice treated with 750 mM of PA or Veh i.p. for 12 hr, before stimulation with LPS. Interestingly, we did not see significant differences in *Tnf, Il6,* or *Il1b* expression at 12 hr of exposure with PA (*Figure 4F*), which suggests that circulating immune cells of these mice is not in a primed state at these time points prior to LPS injection. These data suggest PA induces trained immunity, and not priming; however, the time point of initial inflammation induced by PA remains unknown.

As mentioned previously, canonical inducers of trained immunity (e.g. BCG or β-glucan) induce long-lived enhanced innate immune responses to secondary inflammatory stimuli (*Kaufmann et al., 2018*, *Netea et al., 2020*). Thus, we hypothesized that exposure to a PA bolus would enhance disease severity and mortality in mice, and that this phenotype would persist even after mice were rested from PA injections for 1 week. We injected age matched (6–8 weeks), female BALB/c mice fed SC with a vehicle solution (Veh→SC) or PA (750 mM; PA→SC) i.p. once a day for 9 days and then rested the mice for 1 week. When challenged with systemic LPS, PA→SC showed an increase in disease severity and mortality compared to Veh→SC mice (*Figure 4G and H*), indicating that PA alone can induce long-lived trained immunity that increases susceptibility to inflammatory disease. Importantly, the difference between Veh→SC and PA→SC survival was not significant (*Figure 4H*), suggesting PA is not the sole driver of the enhanced mortality we see in KD.

Lastly, the most commonly studied models for inducing trained immunity are immunization with BCG or stimulation with β-glucan, and they have been shown to protect mice from systemic *Candida albicans* infection via lymphocyte-independent immunological reprogramming that leads to decreased kidney fungal burden (*Kleinnijenhuis et al., 2012*). Therefore, we next tested if PA treatment induces lymphocyte-independent clearance of *C. albicans* infection. For these experiments, *Rag1* knockout (*Rag1*$^{-/-}$) mice were treated with a vehicle or PA solution for 12 hr and subsequently infected intravenously (i.v.) with $2 \times 10^6$ *C. albicans*. In accordance with canonical trained immunity models, mice treated with PA for 12 hr showed a significant decrease in kidney fungal burden compared to Veh mice, 24 hr post-infection (*Figure 4I*). These are the first data to suggest PA enhances innate immune clearance of *C. albicans* in vivo.

## OA reverses enhanced disease severity in WD- and KD-fed mice

We have reported here that diversion of ceramide synthesis reverses the PA-dependent hyper-inflammatory response to LPS in macrophages in vitro (*Figure 3K–M*). Interestingly, OA (C18:1) is a mono-unsaturated fatty acid naturally found in animal fats and vegetable oils, and in the presence of PA, diverts lipid metabolism away from ceramide production (*Palomer et al., 2018*; *Listenberger et al., 2003*). Considering OA and PA are the most prevalent fatty acids found in the human diet and in human serum (*Palomer et al., 2018*), we wanted to test if OA diversion of ceramide synthesis could reverse the PA-dependent hyper-inflammatory response to LPS in macrophages. Thus, we treated BMDMs with OA (0.2 mM), PA (0.5 mM), or OA and PA together for 12 hr and then with LPS. We found that macrophages simultaneously pre-treated with PA and OA produced significantly lower levels of TNF, IL-6, and IL-1β following subsequent LPS exposure, compared to BMDMs pre-treated with only PA prior to LPS stimulation (*Figure 5A–C*). These data reveal OA-dependent depletion of intracellular ceramides neutralizes the PA-dependent hyper-inflammatory response to LPS in macrophages.

Considering this, we next wanted to know if i.p. injections of OA in KD-fed mice would mitigate enriched dietary SFA-associated disease severity and mortality. Thus, we fed age-matched (6–8 weeks) female BALB/c mice SC or KD for 2 weeks and injected them i.p. with 300 mM OA or Veh once per day for the final 3 days of feeding. We then injected LPS i.p. and measured hypothermia and survival. Veh-injected KD-fed mice showed significant and prolonged hypothermia starting at 8 hr p.i. compared to SC-fed mice (*Figure 5D*). In accordance with these findings, KD-fed mice displayed significantly enhanced mortality by 24 hr p.i. compared to 100% survival of SC-fed mice (*Figure 5E*). Strikingly, for KD-fed mice injected with 300 mM OA prior to LPS treatment, there was minimal temperature loss comparable to SC-fed mice, and 100% survival (*Figure 5D and E*). Together, these data show systemic

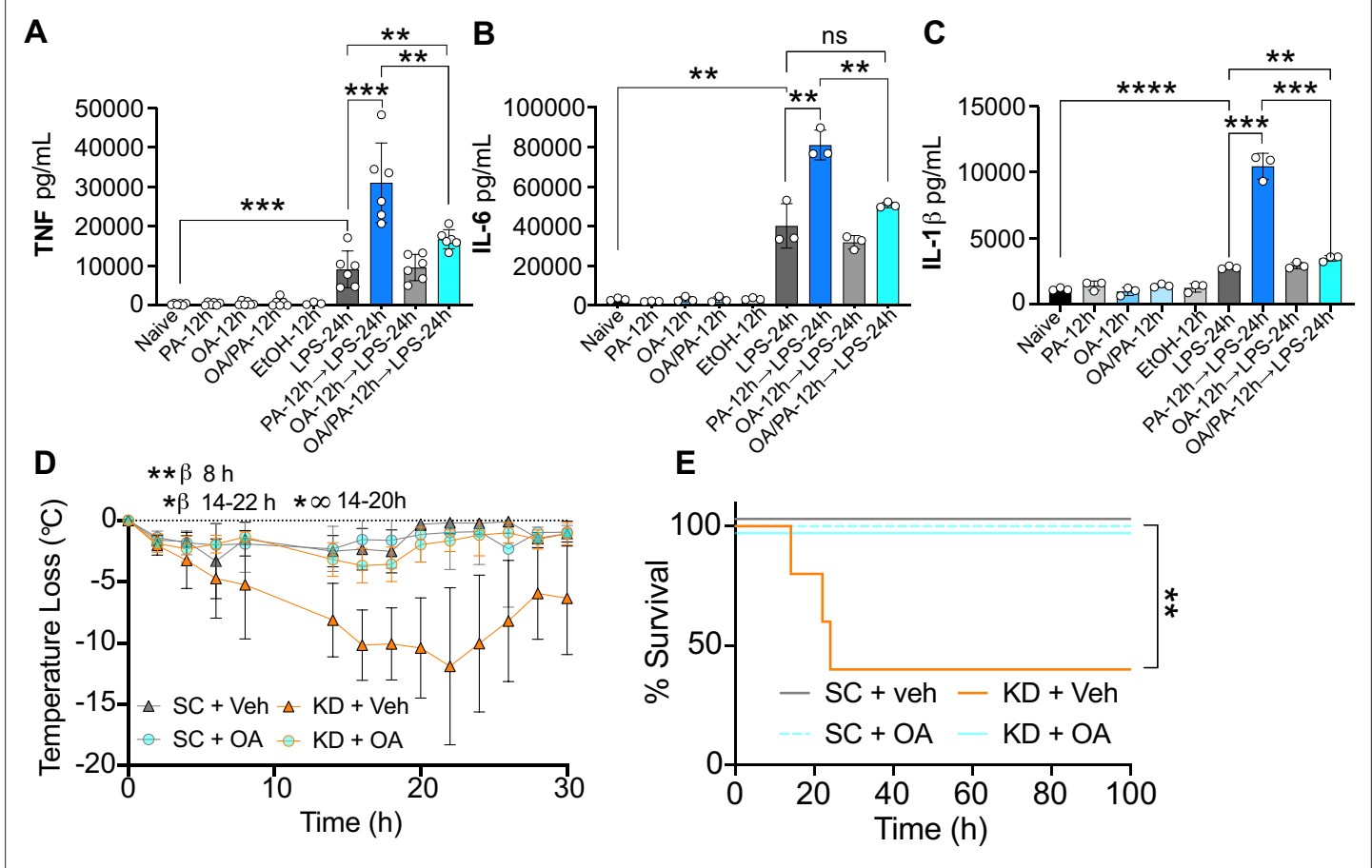

**Figure 5.** Oleic acid (OA) reverses palmitic acid (PA)-dependent hyper-inflammation in response to lipopolysaccharide (LPS) in vitro, and PA-dependent enhanced endotoxemia disease severity in vivo. Primary bone marrow-derived macrophages (BMDMs) were isolated from age-matched (6–8 weeks) C57BL/6 female and male mice. BMDMs were plated at $1 \times 10^6$ cells/mL and treated with either media (Ctrl), LPS (10 ng/mL) for 24 hr, PA (PA stock diluted in 0.83% EtOH; 0.5 mM PA conjugated to 2% bovine serum albumin) for 12 hr, or OA (200 μM; diluted in endotoxin-free water). Controls for all treatments are shown next to experimental groups treated additionally with LPS (10 ng/mL) for 24 hr. Supernatants were assessed via ELISA for (**A**) TNF, (**B**) IL-6, and (**C**) IL-1β secretion. Age-matched (6–8 weeks) female BALB/c mice were fed standard chow (SC) or ketonic diet (KD) for 2 weeks and injected intraperitoneal with 7 mg/kg LPS. (**D**) Temperature loss and (**E**) survival were monitored every 2 hr. For (**A–C**), experiments were run three times and data are representative of (**A**) two experiments and (**B, C**) one experiment. For all plates, all treatments were performed in triplicate, and a student's t-test was used for statistical significance. For (**D**), a Mann Whitney test was used for pairwise comparisons. For (**E**), a log-rank Mantel-Cox test was used for survival curve comparison. For (**D, E**), experiments were run three times, and data are representative of one experiment, n=5 mice/group. β symbols indicate KD + Veh vs KD + OA significance, and ∞ symbols indicate KD + Veh vs. SC + Veh. For all panels, *, p<0.05; **, p<0.01; ***, p<0.001; ****, p<0.0001. Error bars shown mean ± SD.

The online version of this article includes the following source data for figure 5:

**Source data 1.** Data and statistics for graphs depicted in *Figure 5A–E*.

OA can abrogate KD-dependent hypothermia and survival defect in response to LPS in mice-fed diets enriched solely in SFA and highlight the fascinating plasticity of dietary fatty acid reprogramming of innate immune cell populations and disease dynamics.

## Discussion

In this study, we showed that mice-fed diets enriched in SFA exhibit hyper-inflammation during endotoxemia and poorer outcomes, compared with mice fed a standard low-SFA diet, independent of the diet-associated microbiome, ketosis, and the impact of each diet on LPS-induced hypoglycemia (*Figure 1*; *Figure 1—figure supplement 1*). Strikingly, we found that before LPS treatment, healthy mice fed a diet solely enriched in SFAs (KD) displayed significant expansion of HSCs, including MPPs,

and harbored BMDMs, BM monocytes, and splenocytes that were not inherently more inflamed, but when challenged with LPS exhibited increased production of inflammatory cytokines (*Figure 2*; *Figure 2—figure supplement 1*, *Figure 2—figure supplement 2*). Since (*Netea et al., 2020*) we did not confer the hyper-inflammatory phenotype in BMDMs, BMMs, and splenocytes with WD, but only from KD-fed mice, and (*Kleinnijenhuis et al., 2012*) the KD is only enriched in SFAs and contains no sucrose, allowing us to ask questions specifically about SFAs, we chose to focus on the KD for the remainder of the study.

Considering the immunogenic properties of some dietary SFAs enriched in the KD, and that excess dietary SFAs are found circulating throughout the blood and peripheral tissues, we used lipidomics to identify dietary SFAs that may be directly reprogramming innate immune cells to respond more intensely to secondary inflammatory stimuli. Our study identified enriched PA (C16:0; PA) and PA-associated fatty acids in the blood of KD-fed mice (*Figure 3*; *Figure 3—figure supplement 1*). And, when we treated macrophages with physiologically relevant concentrations of PA, we found that PA alone induces a hyper-inflammatory response to secondary challenge with LPS (*Figure 3*; *Figure 3—figure supplement 2*). This enhanced production of inflammatory cytokines in response to secondary heterologous stimuli has been shown in previous models of innate immune memory, specifically trained immunity (*Saeed et al., 2014*; *Christ et al., 2018*; *Bekkering et al., 2014*). Furthermore, our data suggests PA induces trained immunity by showing that circulating inflammatory levels in PA-injected mice were not upregulated or in a primed state prior to LPS stimulation in vivo (*Figure 4F*), and PA-associated enhanced endotoxemia severity and mortality are still shown in mice rested for 7 days post-PA exposure (*Figure 4G–H*). Importantly, we have not fully defined the initial inflammatory response to PA in our model, thus our data only suggests trained immunity is induced by PA exposure. However, we do find that PA exacerbates the acute phase of endotoxin challenge and correlates with increased mortality but also enhances resistance to infection independent of mature lymphocytes (*Figure 4*). Together, our data concludes PA exposure can lead to hallmark phenotypes associated with canonical trained immunity models in vitro and in vivo.

Interestingly, the in vivo blood expression of cytokines for KD-fed mice following endotoxin challenge is mild in comparison to the cytokine secretion we show for BM monocytes, splenocytes, and BMDMs isolated from KD-fed mice treated with LPS ex vivo (*Figure 1*; *Figure 2*; *Supplementary file 2*). The media used for culturing and treating BM monocytes and splenocytes ex vivo with LPS contained a high-glucose concentration (4.5 g/L; 25 mM). However, high-glucose media does not alter TNF, IL-6, or IL-1β secretion, or mitochondrial metabolic activity, in WT BMDMs treated with LPS following 7 days of differentiation in high-glucose media (*Ayala et al., 2019*). Additionally, in these studies, metabolic adaptation likely takes place within 48 hr for BMDMs cultured in high-glucose media (*Ayala et al., 2019*); thus, we suggest it is unlikely that high glucose contributed to the significant augmentation of LPS-induced TNF and IL-6 secretion for BMDMs from KD-fed mice compared to controls, following 7 days of differentiation in high-glucose media prior to LPS challenge. However, further studies on the metabolic flexibility of the SC- and KD-BMDMs will be required to answer this question directly.

Additionally, we have previously shown that WD-induced weight gain does not correlate with enhanced endotoxemia severity and mortality in conventional mice (*Napier et al., 2019*). This is important to address because of the "obesity paradox" that describes the diversity in sepsis severity and mortality exhibited within the obese patient population, with some studies showing that obesity may even be protective in certain disease contexts (*Ng and Eikermann, 2017*). Humans on an animal-based KD that contains 76% fat with 30% SFA content, and 10% carbohydrates, experience ketosis within 1–2 weeks characterized by a three- to fourfold elevation in blood BHB levels and exhibit greater energy expenditure and weight loss compared to humans on a low-fat, plant-based diet that contains 10% fat and 75% carbohydrates (*Hall et al., 2021*). Likewise, KD-fed mice do not gain weight but show enhanced energy expenditure after 5 weeks of diet administration, and a trend toward weight loss during 9 weeks of diet exposure, compared to mice fed an SC diet (*Kennedy et al., 2007*). Thus, neither weight gain nor the obesity paradox is the confounding feature for the data we present here showing that both KD and dietary PA mediate innate immune memory in vivo during endotoxemia.

Furthermore, the metabolism of dietary SFAs is a key element of immune system function, and metabolic intermediates enhanced by SFAs and PA alone, such as ceramide, serve as signaling lipids

in diseases of inflammation (*Galadari et al., 2013*). Mechanistically, we show that inhibiting ceramide synthesis or diverting metabolism away from ceramide synthesis using OA protects macrophages from PA-induced trained immunity, suggesting that dietary intervention may help regulate inflammatory dysregulation during disease (*Figure 5*). And, to complement our in vitro mechanistic findings we show that three single i.p. injections of OA prior to endotoxin stimulation protects KD-fed mice from enhanced disease severity and mortality (*Figure 5*).

Our findings align with the growing body of evidence indicating that trained immunity is a double-edged sword, where the phenomenon can be beneficial for resistance to infection but detrimental in the context of diseases exacerbated by systemic inflammation (*DiNardo et al., 2021*). Specifically, we show that PA-induced memory is beneficial in that it promotes clearance of *C. albicans* infection in the kidneys of *Rag1*$^{-/-}$ mice (*Figure 4I*). In stark contrast, PA-induced memory is detrimental in the context of endotoxemia, a disease driven by organ damage due to acute hyper-inflammation (*Beutler et al., 1985*; *Mohler et al., 1993*; *Cunningham et al., 2002*; *Chen et al., 2012*; *Zhong et al., 2016*; *Figure 4G and H*). Furthermore, it is known that trained immunity is a key feature of BCG vaccination, which has been shown to enhance resistance to infections, and is a possible mechanism that drives increased resistance to severe COVID-19 in the BCG-vaccinated population (*Netea et al., 2016*; *Escobar et al., 2020*). Thus, future research in understanding the plasticity of the SFA- and PA-regulated immune memory responses, enhanced pathogen clearance, and the mechanisms that drive this phenomenon, will be of interest to the larger medical community.

Mechanistically, it is appreciated that PA is not acting as a ligand for the pattern recognition receptor TLR4; however, the presence of TLR4 (independent of TLR4 signaling capability) is required for PA-dependent inflammation (*Lancaster et al., 2018*). Our data and others contribute to the growing evidence that PA is inducing cell intrinsic stress through alterations in metabolism. The crosstalk between glycolytic and oxidative metabolism, and epigenetics, is crucial for trained immunity in human monocytes, and metabolic intermediates of the TCA cycle directly modify histone methylation patterns associated with proinflammatory cytokines upregulated in trained immunity (*Saeed et al., 2014*; *Arts et al., 2016*; *Ryan and O'Neill, 2020*). While ceramides are known to modify histone acetylation and DNA methylation patterns (*Silva et al., 2022*), the interplay between ceramide metabolism and epigenetics within innate immune cells has not been explored. Though we have shown that PA-dependent ceramide production leads to innate immune memory, the impact of these alterations on the epigenome remains unknown. Therefore, the influence of ceramide metabolism on epigenetics will be important to consider in future trained immunity studies where PA serves as the primary stimulus.

Interestingly, we find here that immunoparalysis, which is associated with a prolonged septic response and is enhanced in patients with poorer outcomes, is greater in mice-fed diets enriched in SFAs (*Figure 1*; *Gogos et al., 2000*; *van Dissel et al., 1998*). However, we found that this SFA-dependent enhanced immunoparalysis is abrogated in GF mice, suggesting, for the first time, that the microbial species within the SFA-fed mice may be regulating the late immunoparalytic phase of endotoxin shock. Considering the clinical correlation of immunoparalysis and increased sepsis mortality, it will be imperative to explore the identity of the SFA-dependent microbiome and the host/microbe mechanisms that drive sepsis-associated immunoparalysis.

Importantly, previous seminal studies concluded that mice treated with antibodies to the TNF receptor and challenged with systemic LPS increased survival from 0% to nearly 100%, suggesting that acute inflammation driven by TNF is responsible for endotoxemia-related mortality (*Beutler et al., 1985*; *Mohler et al., 1993*). Furthermore, it has been shown that TNF is required for acute renal failure (*Cunningham et al., 2002*), lung injury (*Chen et al., 2012*), and liver damage (*Zhong et al., 2016*) during LPS challenge. These data show that acute inflammation, specifically the bioactivity of TNF, drives endotoxemia mortality and organ damage in conventional mice. It has also been shown that acute inflammation, specifically TNF production, is a driver of endotoxemia in GF mice (*Souza et al., 2004*). Thus, although our conventional mice show increased immunoparalysis, we suggest that early acute systemic inflammation is the driver of disease severity and mortality in both our conventional and GF endotoxemia mouse models; however, the data we present here is not sufficient to make this conclusion.

In conclusion, this unappreciated role of dietary SFAs, specifically PA, may provide insight into the long-lasting immune reprogramming associated with a high-SFA fed population, and lends insight

into the complexity of nutritional immunoregulation. Considering the results in this study, we suggest the potential for SFAs such as PA to directly impact innate immune metabolism and epigenetics associated with inflammatory pathways. Thus, our findings are paramount not only for potential dietary interventions, but also treatment of inflammatory diseases exacerbated by metabolic dysfunction in humans.

## Materials and methods

### Cell lines and reagents

RAW 264.7 macrophages (from ATCC), CASP-1KO BMDMs, BMDMs, and BMMs were maintained in Dulbecco's Modified Eagle Medium (DMEM; Gibco) containing L-glutamine, sodium pyruvate, and high glucose supplemented with 10% heat-inactivated fetal bovine serum (FBS; GE Healthcare, SH3039603). BMDMs were also supplemented with 10% macrophage colony-stimulating factor (M-CSF; M-CSF-conditioned media was collected from National Institutes of Health (NIH) 3T3 cells expressing M-CSF, generously provided by Denise Monack at Stanford University).

### Generation of BMDMs, BMMs, and splenocytes

BMDMs and BMMs were harvested from the femurs and tibias of age-matched (6–8 weeks) $CO_2$-euthanized female BALB/c mice or male and female C57BL/6 J mice. BMDM media was supplemented with 10% M-CSF for differentiation, cells were seeded at $5\times10^6$ in petri dishes and cultured for 6 days, collected with cold PBS, and frozen in 90% FBS and 10% dimethyl sulfoxide (DMSO) in liquid nitrogen for later use. BMMs were isolated from BMDM fraction using EasySep Mouse Monocyte Isolation Kit (STEMCELL). Spleens were harvested from age-matched (6–8 weeks) $CO_2$-euthanized female BALB/c mice, tissue was disrupted using the end of a syringe plunger on a 70-µm cell strainer and rinsed with FACS buffer (PBS + 2 mM EDTA). Cells were subjected to red blood cell lysis with RBC lysing buffer (Sigma) followed by neutralization in FACS buffer.

### Treatments

After thawing and culturing for 5 days, BMDMs were pelleted and resuspended in DMEM containing 5% FBS, 2% endotoxin- and fatty acid-free BSA (Proliant Biologicals) and 10% M-CSF. Cells were seeded at $2.5\times10^5$ cells/well in 24-well tissue culture plates, treated with EtOH (1.69%, or 0.83%) 10 ng/mL LPS (Ultrapure LPS, *Escherichia coli* 0111:B4, Invivogen), 500 µM or 1 mM PA (Sigma-Aldrich, PHR1120), 10 uM FB1 (Sigma-Aldrich, F1147), or 200 µM OA (Sigma-Aldrich, O7501) and incubated at 37°C and 5% $CO_2$ for 12 or 24 hr. Next, cells were treated with an additional 10 ng/mL LPS and incubated an additional 12 or 24 hr. RAW 264.7 macrophages were thawed and cultured for 3–5 days, pelleted and resuspended in DMEM containing 5% FBS and 2% endotoxin- and fatty acid-free BSA, and treated identical to BMDM treatments. BMMs were seeded immediately after harvesting at $4\times10^{\wedge}5$ cells/well in 96-well V-bottom plates in DMEM containing 10% FBS and treated with LPS for 2 or 24 hr. Splenocytes were seeded immediately after harvesting at $1\times10^5$ cells/well in 96-well V-bottom plates in RPMI media with L-glutamine (Cytiva) containing 10% FBS and treated with LPS for 2 or 24 hr. BMDMs for ex vivo treatments were isolated as described above, plated at $2.5\times10^5$ cells/well in 24-well plates, and stimulated with 10 ng/mL LPS after 12 hr of adherence. For all treatments, supernatant was removed for ELISA analysis, and cells were lysed with TRIzol (ThermoFisher), flash-frozen in liquid nitrogen, and stored at –80°C until qRT-PCR analysis. For all plates, all treatments were performed in triplicate.

### Flow cytometry

Modified panel using combined methods from Kaufmann et al., Nowlan et al., and Vasquez et al. Red blood cells were lysed in BM cells using RBC lysis buffer (Biolegend). BM cells ($3\times10^6$ cells) were stained with viability stain Live/Dead Fixable Aqua (ThermoFisher) at the concentration of 1:200 for 30 min at 4°C. Next, cells were washed with FACS buffer (PBS supplemented with 0.5% BSA; Proliant Biologicals, fatty acid free), and incubated with anti-CD16/32 (clone 93, BioLegend) at a concentration of 1:100 in FACS buffer for 10 min at 4°C. The following antibodies were then used for staining HSCs, and MPPs: anti-Ter-110, anti-CD11b (clone M1/70), anti-CD5 (clone 53–7.3), anti-CD4 (clone RM4-5), anti-CD8a (clone 53–6.7), anti-CD45R (clone RA3-6B2), and anti-Ly6G/C (clone RB6-8C5),

all biotin-conjugated (all BD Bioscience), were added at a concentration of 1:100 for 30 min at 4°C and washed with FACS buffer. Streptavidin-APC-Cy7 (eBioscience), anti-CD150-eFluor450 (clone Q38-480, eBioscience), anti-CD48-PerCPeFluor710 (BD Bioscience), anti-Flt3-PE (clone A2F10.1, BD Bioscience), anti-CD34-PEDazzle 594 (clone HM34, BioLegend), anti-CD27-PE-Cy7 (eBioscience), and anti-CD201-APC (eBioscience) were added all at a concentration of 1:100 for 20 min at 4°C. All cells were then washed with FACS buffer before and after incubation in 1% paraformaldehyde for 30 min at 4°C. Cells were acquired on BD flow cytometer (FACSymphony A1 Cell Analyzer) with FACSDiva Software. Analyses were performed using FlowJo software v.10.1. The DownSample version 3.3.1 plugin was used to standardize events for each sample after populations were gated.

## Lactate dehydrogenase assays

BMDMs were cultured as stated above with culture media, PA, or ethanol in 96-well tissue-culture plates at a concentration of $5 \times 10^4$ cells/well and incubated for 12 hr. Cells were treated with PBS or 10 ng/mL LPS in a phenol-red-free Optimem media (ThermoFisher) and incubated an additional 12 or 24 hr. Supernatants were collected at the specified time points with LDH release quantified with a CytoTox96 Non-Radioactive Cytotoxicity Assay (Promega). Cytotoxicity was measured per well as a percentage of max LDH release, with background media-only LDH release subtracted. For all plates, all treatments were performed in triplicate.

## Measurement of cell viability

Cell viability was determined by 0.4% Trypan Blue dye exclusion test executed by a TC20 Automated Cell Counter (Bio-Rad).

## Blood RNA extraction and real-time qPCR

Mice were treated with PBS or LPS, and at specified time points 10–20 µL of blood was collected from the tail vein, transferred into 50 µL of RNALater (ThermoFisher Scientific), and frozen at –80°C. RNA extractions were performed using RNeasy Mini Kit (Qiagen), and cDNA was synthesized from RNA samples using SuperScript III First-Strand synthesis system (Invitrogen). Gene-specific primers were used to amplify transcripts using SsoAdvanced Universal SYBR Green Supermix (Bio-Rad). A complete list of all primers used, including the names and sequences, is supplied as *Supplementary file 2*.

## Enzyme-linked immunosorbent assay

TNF, IL-6, and IL-1β concentrations in mice serum were measured and analyzed using TNF, IL-6, and IL-1β Mouse ELISA kits (ThermoFisher Scientific), according to the manufacturer's instructions. Absorbances were measured at a wavelength of 450 nm using a microplate reader (BioTek Synergy HTX). Values below the limit of detection (LOD) of the ELISA were imputed with LOD divided by 2 (LOD/2) values.

## LPS-induced endotoxemia model

Age-matched (6–8 weeks) female BALB/c mice were anesthetized with isoflurane and injected subcutaneously with ID transponders (Bio Medic Data Systems). 2 weeks post diet change, and 1 week post ID transponder injection, mice were stimulated with a single injection of 6–10 mg/kg LPS reconstituted in endotoxin-free LAL reagent water (Invivogen) and diluted in PBS for a total volume of 200 µL. Control mice received corresponding volumes of PBS. Progression of disease was monitored every 2 hr after LPS injection for clinical signs of endotoxin shock based on weight, coat and eyes appearance, level of consciousness, and locomotor activity. Age-matched (20–21 weeks) female C57BL/6 mice were treated as described above, except for their LPS dose (4.5 mg/kg). Temperature was recorded using a DAS-8007 thermo-transponder wand (Bio Medic Data Systems). For PA injections, a solution of 750 mM ethyl palmitate (Millipore Sigma), 1.6% lecithin (Sigma-Aldrich), and 3.3% glycerol was made in endotoxin-free LAL reagent water (Lonza). The lecithin-glycerol-water solution was used as a vehicle, and mice were injected with 200 µL of the vehicle as a control or ethyl palmitate solution to increase serum PA levels. For OA injections, a solution of 300 mM OA (Sigma-Aldrich) was made using the same solution and vehicle described above. Mice were injected i.p. with 200 µL of the vehicle as a control, or OA solution, between 7 and 9 pm for 3 days prior to LPS exposure.

## Mouse diets, glucose, and ketones

Six-week-old female mice were fed soft, irradiated chow (PicoLab Mouse Diet 20, product 5058) and allowed to acclimate to research facility undisturbed for 1 week. Chow was replaced by WD (Envigo, TD.88137), KD (Envigo, TD.180423), or SC (Envigo, TD.08485), and mice were fed ad libitum for 2 weeks before induction of endotoxemia. For KD, food was changed daily. For WD, food was changed every 72 hr. Ketones and blood glucose were measured weekly and immediately prior to LPS injections with blood collected from the tail vein using Blood Ketone and Glucose Testing Meter (Keto-Mojo), or with urine collected on ketone indicator strips (One Earth Health, Ketone Test Strips).

## Statistics analysis

Mann Whitney, Mantel-Cox, and student's t-tests were carried out with GraphPad Prism 9.0 software.

## Ethical approval of animal studies

All animal studies were performed in accordance with NIH guidelines, the Animal Welfare Act, and US federal law. All animal experiments were approved by the Oregon Health and Sciences University (OHSU) Department of Comparative Medicine or Oregon State University (OSU) Animal Program Office and were overseen by the Institutional Care and Use Committee (IACUC) under Protocol IDs #IP00002661 and IP00001903 at OHSU and #5091 at OSU. Conventional animals were housed in a centralized research animal facility certified by OHSU. Conventional 6–10-week-aged female BALB/c mice (Jackson Laboratory 000651) were used for the endotoxemia model, and isolation of BMDMs, BMMs, and splenocytes. GF male and female C57BL/6 mice (Oregon State University; bred in house) between 14- and 23-week-old were used for the GF endotoxemia model. BALB/c $Rag1^{-/-}$ mice between 8 and 24 weeks were infected i.v. with $2 \times 10^6$ CFUs of *C. albicans* SC5314 (ATCC #MYA-2876), and kidney fungal burden was assessed 24 hr post-infection. Kidneys were harvested 24 hr post-infection, and homogenized organs were plated in serial dilutions on Yeast Peptone Dextrose plates to assess fungal burden.

## Lipidomics PCA analysis

Mice on specialized diets were sacrificed at the indicated time points after PBS or LPS treatment with 300–600 µL of blood collected via cardiac puncture into heparinized tubes. Blood samples were centrifuged for 15 min at 2500 rpm at 4°C, and serum was transferred to a new tube before storage at –80°C. Serum samples were analyzed via LC-MS/MS. Lipidomic data sets were scaled using the *scale* function, and PCAs were performed using the *prcomp* function from the stats package in R Version 3.6.2. Visualization of PCAs and biplots was performed with the *fviz_pca_ind* and *fviz_pca_biplot* functions from the factoextra package and with the *ggplot2* package (*Mundt and Weisweiler, 2017*; *Wickham, 2016*). For each diet group, 95% confidence ellipses were plotted around the group mean using the *coord.ellipse* function from the FactoMineR package (*Lê, 2008*). Heatmaps were created using the *pheatmap* package (*Kolde, 2018*).

## Acknowledgements

We would like to thank Ajesh Saini, a student in the Napier Lab, for his contributions in carrying out the ELISA data within this manuscript. This study was supported by National Institute of General Medical Sciences (NIGMS) grant 5R35GM133804-02 to B.A.N.

## Additional information

### Funding

| Funder | Grant reference number | Author |
| --- | --- | --- |
| NIGMS/NIH | R35GM133804 | Brooke A Napier |
| National Institute of General Medical Sciences | | Brooke A Napier |

| Funder | Grant reference number | Author |
|---|---|---|
| US Department of Veterans Affairs | VA CDA2 BLRD 1K2BX004523 | Ruth J Napier |

The funders had no role in study design, data collection and interpretation, or the decision to submit the work for publication.

## Author contributions

Amy L Seufert, Conceptualization, Data curation, Formal analysis, Validation, Investigation, Visualization, Methodology, Writing - original draft, Writing - review and editing; James W Hickman, Data curation, Software, Formal analysis, Validation, Investigation, Methodology, Writing - review and editing; Ste K Traxler, Conceptualization, Data curation, Formal analysis, Investigation, Methodology; Rachael M Peterson, Data curation, Formal analysis, Investigation; Trent A Waugh, Investigation, Methodology; Sydney J Lashley, Data curation, Investigation; Natalia Shulzhenko, Resources, Project administration; Ruth J Napier, Conceptualization, Resources, Data curation, Investigation, Project administration; Brooke A Napier, Conceptualization, Resources, Data curation, Formal analysis, Supervision, Funding acquisition, Investigation, Visualization, Methodology, Writing - original draft, Project administration, Writing - review and editing

## Author ORCIDs

Amy L Seufert http://orcid.org/0000-0001-9777-6793
Brooke A Napier http://orcid.org/0000-0002-6409-4750

## Ethics

This study was performed in strict accordance with the recommendations in the Guide for the Care and Use of Laboratory Animals of the National Institutes of Health. All of the animals were handled according to approved institutional animal care and use committee (IACUC) protocols (#IP00002661 & IP00001903) of Oregon Health & Sciences University and Oregon State University (#5091). All animal experiments were approved by the Oregon Health and Sciences University (OHSU) Department of Comparative Medicine or Oregon State University (OSU) Animal Program Office and were overseen by the Institutional Care and Use Committee (IACUC).

## Decision letter and Author response

Decision letter https://doi.org/10.7554/eLife.76744.sa1
Author response https://doi.org/10.7554/eLife.76744.sa2

# Additional files

## Supplementary files

- Supplementary file 1. Diet compositions (values represent percentage of total kcal).
- Supplementary file 2. List of primers used in this study.
- Transparent reporting form

## Data availability

All data generated or analyzed during this study are included in the manuscript and supporting file.

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
