## [Editor Report]

This fundamental paper in mice convincingly demonstrates that a Western-type diet and the more extreme ketogenic diet for 2 weeks enhance monocyte-driven immune responsiveness. This leads to a deadly hyper-inflammatory state in the mice in response to an endotoxin challenge in vivo and enhances the clearance of pathogens. The paper is of interest to immunologists, infectious disease specialists, and nutritionists.

---

## [Decision Letter]

**Decision letter after peer review:**

[Editors’ note: the authors submitted for reconsideration following the decision after peer review. What follows is the decision letter after the first round of review.]

Thank you for submitting the paper "Dietary palmitic acid induces innate immune memory via ceramide production that enhances severity of acute septic shock and clearance of infection" for consideration by *eLife*. Your article has been reviewed by 2 peer reviewers, and the evaluation has been overseen by a Reviewing Editor and a Senior Editor. The following individual involved in review of your submission has agreed to reveal their identity: Maziar Divangahi (Reviewer #2).

Comments to the Authors:

We are sorry to say that, after consultation with the reviewers, we have decided that this work will not be considered further for publication by *eLife*.

Specifically, the shortcomings of the paper, which are detailed in the reviews below, are such that the work required for an acceptable revision cannot be delivered within the 8 weeks that we allow.

*Reviewer #1 (Recommendations for the authors):*

1. Presumably Figure 1 panels J-N are data stemming from both male and female mice. Although this is great for sexual diversity and representation in scientific studies. This decision is weakened by 2 issues (1) the sample sizes are very small, and therefore don't allow the authors to investigate sex specific difference in the mice's response to diet and LPS. (2) In humans males and females show hugely different responses to endotoxemia challenge [PMID: 17452928]. According to a fairly recent PLoS ONE publication [PMID: 27631979] these sex differential responses are also present in mice, where male mice have a greater inflammatory response and have more severe outcomes. Since readers aren't given any insight into the sex distribution of the mice in each group it is impossible to decide if the data are skewed/biased. In order to improve the robustness of the authors findings, these studies should be repeated with mice sharing a single sex, or be performed with greater numbers (n=>12) with equal distribution of sexes in each dietary group.

2. Figure 4 D – F shows the OA for 12h followed by LPS for 24h induces the greatest levels of cytokines. In fact, what's worrying with these data is that the effect of "PA-12h◊LPS-24h" vs "LPS-24h" is not recapitulated from the previous results shown in Figure 4 panels A – C; at least the authors don't display the results of statistical tests for panels D – F, however comparing the cytokine levels it's clear that the PA treatment didn't generate the same cytokine levels. How do the authors explain these results? Also why isn't the highly cytokine inducing potential of OA not described in the Results section? In fact what is shown in Panels D – F is that PA treatment reverses the inflammation induced by OA, rather than the other way around that the authors claim to be the case. These experiments should be repeated, showing a significant change from "LPS-24h" to "PA-12hLPS-24h" as in panels Figure 4 A – C. The authors should also address the effect of OA on LPS responsiveness.

I would like to politely highlight a concern of mine, the authors appear to have purposefully chosen their color scheming to throw the eye off from noticing the effect of "OA-12hLPS-24h". Notice how in panels A – C "PA-12hLPS24h" is colored in blue, while in panels D – F "PA-12h◊LPS-24h" is colored in grey and "OA-12hLPS-24h" is colored in blue. Skimming of the figures, one could easily mistake that both blue columns represent "PA-12hLPS-24h", completely overlooking that OA is inducing high levels of cytokines and that the authors failed to reproduce their own results. The authors should use the same color coding for both sets of figures.

3. Dietary intervention studies are difficult, and often difficult decisions have to be made when designing your studies. One of these difficult decisions is how to design the macronutrient contributions to the total caloric intake of the mice. A choice could be made to, for example, to fix the percentage or mg of protein to the caloric total. In this way it is possible control for the effect that altering protein availability will have on LPS challenge. For example protein intake effects the survivaly of critically ill patients – some with sepsis [PMID: 29709380]. Can the authors justify their decision for the nutrient composition used in their study?

4. In basic/functional science it's important to make reductionist groupings for the sake of clear comparisons (also for financial reasons). However, what may give you the clearest separation in your analysis may impact the translatability of the data to the broader, medical context. Here the authors have decided to compare standard chow diets (SC) with the ketogenic diet (KD), as opposed to the potentially more interesting (from the epidemiological perspective at least) Western-diet (WD). It is unsurprising that eliminating most carbohydrates from ones diet will result in gross changes in systemic metabolism and circulating lipid profiles. However how common are septic events in people practicing the ketogenic diet? For a lay audience this may convey the message that ketogenic diet – per se – is a major contributor to septic death. Naturally the authors tie these results to dietary palmitic acid, where supplementation alone is able to confer these negative effects. However I think the manuscript would benefit from a small sentence in the results prior to Figure 2 outlining why they chose to focus on KD rather than WD.

5. The description of the methods are rather poor, but also there appear to be many inconsistencies in the methods themselves. In the legend of Figure 1 the authors use "conventional mice" aged 4-6 weeks for the data shown in panel A – F, while through panel G – N the switch to C57BL/6 mice that are 19-23 weeks old female or 14-23 weeks for male mice; using a unreported mixture of the sexes. Additionally, the younger mice receive 6 mg/kg of LPS, while the older mice receive 50 mg/kg of LPS. Later on in Figure 5, switching back to the younger 4-6 week mice (all female), the authors switch again to 10 mg/kg of LPS. The RAG -/- mice are represent a third age group used in the study (8-9 weeks old). I see this as a major, and limiting aspect of the presented work. If it's within the power of the authors, I would advocate for repeating the mice studies in the paper with a single, standardized protocol. Using mice from the same background, sex, age, and to receive the same injection of LPS.

*Reviewer #2 (Recommendations for the authors):*

In this manuscript entitled "Dietary palmitic acid induces innate immune memory via ceramide production that enhances severity of acute septic shock and clearance of infection" Seufert and colleagues have investigated how saturated fatty acids increase susceptibility of the host in a murine model of LPS-mediated septic shock. They have shown that pretreatment of macrophages with palmitic acid (PA) reprograms macrophages towards hyper-inflammatory phenotype, which was dependent on ceremide. Importantly, depletion of macrophages intracellular ceremide with oleic acid reverse their hyper-inflammatory phenotype. Interestingly, while PA was harmful in the LPS-acute septic shock model, it was beneficial in clearance of *C. albicans* in Rag deficient mice lacking both B and T cells. While this is an exciting study, the presented data don't fully support the central hypothesis and the link with trained immunity is currently weak.

1. Training: As the authors described in the result section the difference between priming and training: "priming occurs when the first stimulus enhances transcription of inflammatory genes and doesn't return to basal levels before the secondary stimulation. In contrast, trained immunity occurs when the first stimulus changes transcription of inflammatory genes, the immune status then returns to basal levels, and challenges with homologous or heterologous stimulus enhances transcription of inflammatory cytokines at much higher levels than those observed during the primary challenge". However, the presented in vitro data are priming (Figure 3 and 4) and not training, as the interval between the first (PA) and secondary (LPS) stimulus was extremely short. Indeed, PA often increased the expression or production of inflammatory cytokines, in which was augmented following LPS stimulation. Epigenomic/transcriptomic approaches can more precisely distinguish between priming and training. I would suggest the authors to move the definition of priming vs training from the result section to the introduction.

2. Microbiome: In Figure 1H-N, by using GF-mice, the authors indicated that SFA-driven enhanced responses to systemic LPS was independent of microbiome. However, the mortality was increased by 50% in SC-GF (Figure 1I) mice compared to SC-WT (Figure 1B) mice. Thus we could argue that the increased mortality in control groups was dependent on microbiome while it was masked in super susceptible WD/KD groups. This needs to be carefully addressed.

3. Cellular sources: In Figure 1, the authors assess the expression of several pro-inflammatory genes as well as an anti-inflammatory gene in blood, which composed of 90% lymphocytes and only 5-10% myeloid cells. Thus it is not clear to this reviewer that how the increased expression of these genes are linked to trained innate immunity. Similarly, in Figure 2, the authors showed that the expression of TNF was increased in splenocytes (>80% lymphocytes) of KD mice after ex-vivo stimulation with LPS. So, without knowing the cellular sources of these cytokines, it is challenging to directly link the expression of inflammatory genes to innate cells. Furthermore, the authors need to provide the level of circulating cytokine by simply using ELISA. Considering the overlapping error bars in evaluating of many gene expressions statistical analysis needs to be double checked.

4. Hematopoietic stem cells: In Figure 2, the authors showed that monocytes isolated from BM of KD-mice expressed higher levels of TNF and IL6 following ex-vivo LPS stimulation. These data indicate that the potential impact of SFAs is on HSCs and progenitor cells. I would highly recommend the authors to consdier two experiments to provide more direct link between SFAs and trained immunity. First, to assess the impact of SFAs or PA on HSCs and myeloid progenitor cells by FACS as it has been done in the BCG model (Kaufmann et al., 2018). Second, to generate BMDM from SFAs and SC mice and then stimulate them with LPS or infect them with *C. albicans*, ex-vivo. As the generation of BMDM will take 5-7 days, any initial imprinting by SFAs can be transmitted from the progenitor cells to macrophages. Thus if there is an alteration in HSCs and progenitor cells, this can provide mor evidence of central training and its long-term effects on immunity to sterilized immunity (LPS) or infections (e.g. *C. albicans*). This will also provide more support for Figure 5H-I.

[Editors’ note: further revisions were suggested prior to acceptance, as described below.]

Thank you for resubmitting your work entitled "Enriched dietary saturated fatty acids induce trained immunity via ceramide production that enhances severity of acute septic shock and clearance of infection" for further consideration by *eLife*. Your revised article has been evaluated by Jos van der Meer (Senior Editor) and a Reviewing Editor.

The manuscript has been improved but there are some remaining issues that need to be addressed, as outlined below:

The authors have wonderfully addressed the comments, and have resubmitted a very strong and thorough study on the role of dietary fatty acids on the induction of in vivo trained immunity. The manuscript is nearly ready for publication, especially considering the novelty and strength of their findings, not to mention the relevance to the field of trained immunity as well as metainflammation. There are however a few final comments on the present manuscript.

The first thing is the so-called 'obesity paradox'. People overweight or with obesity are generally protected from deadly/lethal septicemia when compared to normal or underweight individuals. Two weeks of high-fat feeding may not be sufficient to induce obesity, however, it would be interesting to know if the mice have gained weight. It would be of interest to briefly discuss the clinical conundrum of the 'obesity paradox', especially given the highly lethal phenotype of WD and KD-fed mice in for example Figure 1B.

With sepsis, the general course of the disease is indeed hyper-inflammation leading to organ failure and death, or immune paralysis and increased risk of mortality via secondary infection. Given that your data suggest that the mice indeed enter into the immune-paralyzed state, do you have any data to show whether the mice died from organ failure or secondary infections? Regardless of the phenotype, WD or KD-fed mice are more susceptible to lethal responses to LPS. Germ-free mice similarly show lethality making the effect clearer to being the result of organ failure/damage. Though, the cytokines, especially IL-10 make this distinction even murkier. The germ-free clearly look as though they are dying from hyper-inflammation, while the BALB/c mice are seemingly dying in the immune-paralyzed state; given they aren't germ-free and are susceptible to secondary infection with commensals. How would the authors rationalize this?

Sepsis is a medical term that describes an infection of the blood. In your model, LPS is used to induce systemic inflammation. Although many of the symptoms and effects overlap profoundly with those seen in sepsis, we advise against using the term sepsis to describe your model since no infection is established. Endotoxin challenge, or endotoxin stimulation, is more appropriate/accurate semantics.

Page 17 lines 461 – 463 reading: "Our findings align with the growing body of evidence indicating that trained immunity is a double-edged sword, where the phenomenon can be beneficial for resistance to infection, but detrimental in the context of inflammatory disease." This reads a bit off given that, in fact, your model uses a bacterial-derived stimulus to induce a lethal inflammatory response in WD and KD-'trained' mice. You could hardly call this a beneficial effect of trained immunity.

A technical hurdle for figure 3 is that the mice are on a ketogenic diet, cells are harvested, and then brought into culture with glucose-enriched media. We wonder then how the sudden overabundance of glucose might contribute to the elevated cytokine responsiveness seen in the data presented in these figure panels, as opposed to the ketogenic diet per se giving rise to increased immune responsiveness. It is worth saying that the in-vivo cytokine responses are much milder than what's seen ex vivo.

---

## [Author Response]

[Editors’ note: The authors appealed the original decision. What follows is the authors’ response to the first round of review.]

Reviewer #1 (Recommendations for the authors):1. Presumably Figure 1 panels J-N are data stemming from both male and female mice. Although this is great for sexual diversity and representation in scientific studies. This decision is weakened by 2 issues (1) the sample sizes are very small, and therefore don't allow the authors to investigate sex specific difference in the mice's response to diet and LPS. (2) In humans males and females show hugely different responses to endotoxemia challenge [PMID: 17452928]. According to a fairly recent PLoS ONE publication [PMID: 27631979] these sex differential responses are also present in mice, where male mice have a greater inflammatory response and have more severe outcomes. Since readers aren't given any insight into the sex distribution of the mice in each group it is impossible to decide if the data are skewed/biased. In order to improve the robustness of the authors findings, these studies should be repeated with mice sharing a single sex, or be performed with greater numbers (n=>12) with equal distribution of sexes in each dietary group.

The Reviewer brings up an important point, all studies with endotoxemia in wild-type conventional mice were carried out in 6–8-week female BALB/c mice, as mentioned in the Methods section under “Ethical approval of animal studies” and “endotoxin-induced model of sepsis” sections. This is extremely important to mention more clearly in the results text, because the Reviewer 1 is correct, sexual dimorphism and age differences can have very large effects on LPS treatment outcome. This was not stated clearly enough in the results and now the age, sex, and background of mice have been explicitly stated in each Results and Figure Legend section for each experiment.

2. Figure 4 D – F shows the OA for 12h followed by LPS for 24h induces the greatest levels of cytokines. In fact, what's worrying with these data is that the effect of "PA-12h◊LPS-24h" vs "LPS-24h" is not recapitulated from the previous results shown in Figure 4 panels A – C; at least the authors don't display the results of statistical tests for panels D – F, however comparing the cytokine levels it's clear that the PA treatment didn't generate the same cytokine levels. How do the authors explain these results? Also why isn't the highly cytokine inducing potential of OA not described in the Results section? In fact what is shown in Panels D – F is that PA treatment reverses the inflammation induced by OA, rather than the other way around that the authors claim to be the case. These experiments should be repeated, showing a significant change from "LPS-24h" to "PA-12hLPS-24h" as in panels Figure 4 A – C. The authors should also address the effect of OA on LPS responsiveness.I would like to politely highlight a concern of mine, the authors appear to have purposefully chosen their color scheming to throw the eye off from noticing the effect of "OA-12hLPS-24h". Notice how in panels A – C "PA-12hLPS24h" is colored in blue, while in panels D – F "PA-12h◊LPS-24h" is colored in grey and "OA-12hLPS-24h" is colored in blue. Skimming of the figures, one could easily mistake that both blue columns represent "PA-12hLPS-24h", completely overlooking that OA is inducing high levels of cytokines and that the authors failed to reproduce their own results. The authors should use the same color coding for both sets of figures.

The reviewer brings up an important point, Eguchi *et al.* did use infusions. From their data (Figure 1A), we calculated that after 600mM of i.v. injection (total = 267uL within 14h; 0.2L/min) there was ~420uM absolute PA within the blood. They were using C57BL/6 mice that were 23g on average. Using these results, we extrapolated that one single 200uL injection of a 750mM PA solution within 6–8-week female BALB/c mice (~15-18g) would equate to ~500-1mM of PA within the blood. Considering obese healthy and unhealthy humans vary widely in total PA concentrations in the blood (0.3-4.1 mM) (*1, 2*), we moved forward with these calculations. Considering this, we thank the reviewer for this advice, and we agree that we have not definitively shown we are increasing systemic levels of PA. Thus, we ran a lipidomic analysis of serum from SC-fed mice with Veh or PA for 12 h. We show that a 750 mM i.p. injection of ethyl palmitate enhances free PA levels in the serum to 173-425 μM at 2 h post-injection, which is within the reported range for humans on high-fat diets (0.34.1mM). We have added this new data to Figure S7A of the main manuscript.

Importantly, the concentration in the PA-treated mice is greater than that of the Veh-treated mice, however we believe the value shown is an underestimate of maximum serum PA levels enhanced by i.p. injection, because free PA is known to be packaged into chylomicrons within enterocytes and travel through the circulation with a half-life of less than an hour (*3, 4*). Thus, serum concentrations of free PA are only transiently enhanced by i.p. injection, and is quickly taken up by adipose tissue, skeletal muscle, heart, and liver tissue. These complex lipid transport processes make it difficult to determine maximum concentrations of free PA in the serum.

While all of the details concerning PA circulation following an i.p. injection are unknown, we suggest that this method of “force-feeding” is similar to dietary intake in that uptake of PA into the circulation occurs within the peritoneal space prior to traveling to the blood via the thoracic duct and right lymphatic duct (*5*).

3. Dietary intervention studies are difficult, and often difficult decisions have to be made when designing your studies. One of these difficult decisions is how to design the macronutrient contributions to the total caloric intake of the mice. A choice could be made to, for example, to fix the percentage or mg of protein to the caloric total. In this way it is possible control for the effect that altering protein availability will have on LPS challenge. For example protein intake effects the survivaly of critically ill patients – some with sepsis [PMID: 29709380]. Can the authors justify their decision for the nutrient composition used in their study?

The reviewer brings up an important and nuanced topic in the immunometabolism field. Our diets are created by Envigo, and each diet (SC, WD, and KD) contain the same vitamin mixes, carbohydrate, and fat sources.

We originally conducted experiments in SC and WD to understand the role of high SFA and sucrose on reprogramming innate immune populations (*7*). However, in this study we wanted identify if enriched SFA alone could recapitulate these results. The mice were tolerant to the Envigo KD, so we continued our studies. We specifically state this within the Results section, but considering this important point, we have added these diet details within the Supplemental Table 1.

It is important to note, the KD has depleted protein (9.1% compared to 17% and19% in SC and WD, respectively). Our studies have not ruled out that low protein is involved in the response to systemic LPS; however, we have recapitulated these results in short-term PA-treated mice (fed SC; Figure 4A-F) and long-term PA-treated mice (fed SC; Figure 4G-H) suggesting that regardless of the effect of low protein on this model, enriched SFA is sufficient to drive hyper-sensitivity to systemic LPS.

4. In basic/functional science it's important to make reductionist groupings for the sake of clear comparisons (also for financial reasons). However, what may give you the clearest separation in your analysis may impact the translatability of the data to the broader, medical context. Here the authors have decided to compare standard chow diets (SC) with the ketogenic diet (KD), as opposed to the potentially more interesting (from the epidemiological perspective at least) Western-diet (WD). It is unsurprising that eliminating most carbohydrates from ones diet will result in gross changes in systemic metabolism and circulating lipid profiles. However how common are septic events in people practicing the ketogenic diet? For a lay audience this may convey the message that ketogenic diet – per se – is a major contributor to septic death. Naturally the authors tie these results to dietary palmitic acid, where supplementation alone is able to confer these negative effects. However I think the manuscript would benefit from a small sentence in the results prior to Figure 2 outlining why they chose to focus on KD rather than WD.

We respect the opinion of the reviewer, and suggest the KD was a jumping off point for identifying enriched saturated fatty acids and understanding the more interesting mechanistic data provided by Figures 2-5. We chose the KD diet because it was only enriched in SFA and not glucose, like the WD. We agree with the reviewer that a sentence in the Results prior to Figure 3 outlining why we chose the KD rather than the WD is needed. Thus, we have addressed the reviewers concern and have explained why we chose to focus on the KD diet exclusively in the beginning of the Results section, “Palmitic acid (PA) and PA-associated fatty acids are enriched in the blood of KD-fed mice.” We state, “Considering that the KD is enriched in SFAs and not sucrose, and that KD-fed mice showed distinct HSC alterations and LPS-induced hyper-inflammation in BMDMs, BMMs, and splenocytes treated ex vivo (Figure 2; S2), the subsequent studies were performed exclusively on KD-fed mice.”

5. The description of the methods are rather poor, but also there appear to be many inconsistencies in the methods themselves. In the legend of Figure 1 the authors use "conventional mice" aged 4-6 weeks for the data shown in panel A – F, while through panel G – N the switch to C57BL/6 mice that are 19-23 weeks old female or 14-23 weeks for male mice; using a unreported mixture of the sexes. Additionally, the younger mice receive 6 mg/kg of LPS, while the older mice receive 50 mg/kg of LPS. Later on in Figure 5, switching back to the younger 4-6 week mice (all female), the authors switch again to 10 mg/kg of LPS. The RAG -/- mice are represent a third age group used in the study (8-9 weeks old). I see this as a major, and limiting aspect of the presented work. If it's within the power of the authors, I would advocate for repeating the mice studies in the paper with a single, standardized protocol. Using mice from the same background, sex, age, and to receive the same injection of LPS.

We appreciate this review and suggest that:

1) For the LPS models, mice were all female and aged matched between 6-8 weeks. We are aware of sex differences in the endotoxemia model, which is why we specifically use female mice in our studies (*6, 7*). This is mentioned twice in the methods under the sections “Endotoxin-induced model of sepsis” and “Ethical approval of animal studies”. We have added these specifics of our model to all Results and Figure Legend sections for clarification.

2) For Germ-free models, it is notoriously difficult to breed C57BL/6 germ-free mice. It was inherently difficult to obtain enough mice within the same sex and age to carry out these experiments, however since we have published in this model before with mixed sex and age we were aware that our WD phenotype is robust enough in these backgrounds (*7*). Further, we believe that seeing our robust phenotype independent of age or sex within germ-free mice provides more evidence of the strength of this phenotype. It is important to note that we induce endotoxemia within Germ-free mice with 50mg/kg, instead of 6mg/kg which is used in conventional mice, because this is our reported LD50 for mixed sex Germ-free C57BL/6, as we have published previously in detail (*7*). This difference is due to the presence of the microbiota (*8, 9*) and also germ-free mice have an immature immune system that correlates with a hypo-responsiveness to microbial products (*10-12*). We agree with the reviewer that the ages of the C57BL/6 germ-free mice are significantly older than our conventional 6-8 week mice, thus we confirmed that WD- and KD-fed conventional C57BL/6 female mice aged 20 – 21 weeks old still show enhanced disease severity and mortality in an LPS-induced endotoxemia model, compared to mice.

3) In our preliminary results, we stratified survival during *C. albicans* infection between male and female C57BL/6 and found no notable difference in survival at 40h post IP infection with *Candida albicans* (Author response image 1). However, the data presented in the manuscript on CFU is female kidney burden and we do not have data on fungal burden within male mice. This is an important piece of data that we would like to collect for understanding sex differences in the PA-dependent enhanced resistance to systemic *C. albicans*. We are currently addressing this question within the lab as well as elucidating the cell type and mechanism of PA-dependent enhanced fungal resistance.

**Author response image 1. sa2fig1:** PA treatmentenhances survival in both female and male RAG^-/-^ mice. Age-matched (8-9 wk) RAG^-/-^ mice were injected i.v. with ethyl palmitate (PA, 750mM) or vehicle (Veh) solutions 12 h before *C.albicans* infection. Survival was monitored for 40h post-infection fed SC (Figure S1G-H).

Reviewer #2 (Recommendations for the authors):In this manuscript entitled "Dietary palmitic acid induces innate immune memory via ceramide production that enhances severity of acute septic shock and clearance of infection" Seufert and colleagues have investigated how saturated fatty acids increase susceptibility of the host in a murine model of LPS-mediated septic shock. They have shown that pretreatment of macrophages with palmitic acid (PA) reprograms macrophages towards hyper-inflammatory phenotype, which was dependent on ceremide. Importantly, depletion of macrophages intracellular ceremide with oleic acid reverse their hyper-inflammatory phenotype. Interestingly, while PA was harmful in the LPS-acute septic shock model, it was beneficial in clearance of C. albicans in Rag deficient mice lacking both B and T cells. While this is an exciting study, the presented data don't fully support the central hypothesis and the link with trained immunity is currently weak.1. Training: As the authors described in the result section the difference between priming and training: "priming occurs when the first stimulus enhances transcription of inflammatory genes and doesn't return to basal levels before the secondary stimulation. In contrast, trained immunity occurs when the first stimulus changes transcription of inflammatory genes, the immune status then returns to basal levels, and challenges with homologous or heterologous stimulus enhances transcription of inflammatory cytokines at much higher levels than those observed during the primary challenge". However, the presented in vitro data are priming (Figure 3 and 4) and not training, as the interval between the first (PA) and secondary (LPS) stimulus was extremely short. Indeed, PA often increased the expression or production of inflammatory cytokines, in which was augmented following LPS stimulation. Epigenomic/transcriptomic approaches can more precisely distinguish between priming and training. I would suggest the authors to move the definition of priming vs training from the result section to the introduction.

We thank the reviewer for their thoughtful comments on this topic. We agree, and thus, we have added to Introduction of the manuscript, we describe the difference between priming and trained immunity with the following, “Importantly, trained immunity is induced when a primary inflammatory stimulus changes transcription of inflammatory genes, the immune status returns to basal levels, and challenge with a homologous or heterologous stimulus enhances transcription of inflammatory cytokines at much higher levels than those observed during the primary challenge (*14*). While the dynamics of basal inflammation are not defined in this paper, we show that basal levels of *tnf*, *il-6*, *il-1β* and *il-10* in the blood of mice pre-exposed to PA were comparable to control mice immediately prior to LPS-induced sepsis, indicating that mice were not in a primed state prior to disease. This suggests that the hyper-inflammation and poor disease outcome we show in PA-exposed mice is due to trained immunity, and not priming.”

Importantly, the additional ex vivo experiments this Reviewer suggests in point #4 has addressed this trained immunity phenotype further.

2. Microbiome: In Figure 1H-N, by using GF-mice, the authors indicated that SFA-driven enhanced responses to systemic LPS was independent of microbiome. However, the mortality was increased by 50% in SC-GF (Figure 1I) mice compared to SC-WT (Figure 1B) mice. Thus we could argue that the increased mortality in control groups was dependent on microbiome while it was masked in super susceptible WD/KD groups. This needs to be carefully addressed.

This reviewer brings up a very important point. We have addressed these issues thoroughly and specifically in our previous publication: Napier, et al. *Western diet regulates immune status and the response to LPS-driven sepsis independent of diet-associated microbiome*. PNAS. 2019 (Figure 4A-D). However, we were not clear enough in our text and will change: “Diets enriched in SFAs drive enhanced responses to systemic LPS independent of microbiome” to à “Diets enriched in SFAs drive enhanced responses to systemic LPS independent of diet-associated microbiome”, where appropriate.

3. Cellular sources: In Figure 1, the authors assess the expression of several pro-inflammatory genes as well as an anti-inflammatory gene in blood, which composed of 90% lymphocytes and only 5-10% myeloid cells. Thus it is not clear to this reviewer that how the increased expression of these genes are linked to trained innate immunity. Similarly, in Figure 2, the authors showed that the expression of TNF was increased in splenocytes (>80% lymphocytes) of KD mice after ex-vivo stimulation with LPS. So, without knowing the cellular sources of these cytokines, it is challenging to directly link the expression of inflammatory genes to innate cells. Furthermore, the authors need to provide the level of circulating cytokine by simply using ELISA. Considering the overlapping error bars in evaluating of many gene expressions statistical analysis needs to be double checked.

We agree with the reviewer, typically only 4 and 10% of leukocytes in the blood, in mice and humans respectively, are considered myeloid cells. Additionally, within the first 24 hours after systemic LPS exposure in humans (in vivo) the % of classical monocytes (expressing inflammatory cytokines TNF and IL-1b; CD14^hi^CD16^-^) increases in the blood, and to a lesser degree nonclassical monocytes (expressing TNF, IL-1b, IL-6, and IL-8; CD14^dim^CD16^hi^) increase in the blood (*15*). These systemic pools of monocytes are then recruited to peripheral tissues whereby they differentiate into tissue resident macrophages and DCs [Reviewed here Ginhoux, 2014, Nature Reviews Immunology].

In mice, after exposure to systemic LPS, inflammatory bone marrow monocytes (CD11b^+^Ly6C^hi^ – mouse Ly6C^hi^ monocytes are equivalent to human CD14^hi^ monocytes), lung monocytes, and splenic monocytes [Reviewed here: Teh *et al.*, Frontiers in Immunology, 2019] are recruited to peripheral tissues through the blood. These blood monocyte populations are typically transient, and decrease after inflammation resolves.

Further, we have specifically seen that 10h post-LPS inject, inflammatory monocytes (CD11b^+^CD115^+^) are increased within the blood in SC mice (*7*). Other labs have shown, inflammatory (CD11b^+^CD115^+^Ly6^hi^) monocytes are recruited to peripheral tissues, like the liver, 24h post-LPS treatment [Reviewed here Ginhoux, 2014, Nature Reviews Immunology]. This suggests there is a transitory period that inflammatory monocytes producing cytokines are traveling within the blood, and may be responsible for this acute increase in cytokine expression 5-20h after LPS induction (Figure 1).

This topic is of immense interest to our lab, and we are currently working to identify the leukocytes responsible for increased expression of inflammatory cytokines within the blood during endotoxemia between diets.

To address the last comments, we assessed cytokine secretion within the mice during LPS through 10uL tail vein bleeding – allowing us to track the progression of disease and expression of cytokines throughout disease within each mouse. This volume of blood does not allow for ELISA analysis and would require sacrificing mice. Lastly, we have provided all statistical analysis of our cytokine expression data within the reference source data.

4. Hematopoietic stem cells: In Figure 2, the authors showed that monocytes isolated from BM of KD-mice expressed higher levels of TNF and IL6 following ex-vivo LPS stimulation. These data indicate that the potential impact of SFAs is on HSCs and progenitor cells. I would highly recommend the authors to consdier two experiments to provide more direct link between SFAs and trained immunity. First, to assess the impact of SFAs or PA on HSCs and myeloid progenitor cells by FACS as it has been done in the BCG model (Kaufmann et al., 2018). Second, to generate BMDM from SFAs and SC mice and then stimulate them with LPS or infect them with C. albicans, ex-vivo. As the generation of BMDM will take 5-7 days, any initial imprinting by SFAs can be transmitted from the progenitor cells to macrophages. Thus if there is an alteration in HSCs and progenitor cells, this can provide mor evidence of central training and its long-term effects on immunity to sterilized immunity (LPS) or infections (e.g. C. albicans). This will also provide more support for Figure 5H-I.

We want to thank the reviewer for these suggestions, and we believe that these suggested experiments have bolstered our results significantly. As the reviewer suggested, we have used a modified version of the Kaufmann *et al.* 2018 FACs panel (Figure 1) to identify the proportions of long-term HSCs, short-term HSCs, and multipotent progenitors after 2 weeks of diet exposure (SC, WD, and KD). Further, we quantified MPP3s, though we had too low MPP4s to conclude results. We suggest the identity of MPP4s may be different in BALB/c mice (used here) compared to C57BL/6 mice (used by Kaufmann *et* al). Figure 2 now shows the analysis of LT-HSCs, ST-HSCs, and MPPs sub-types from the bone marrow of SC-, WD-, and KD-fed mice.

Supplemental Figure 2A shows quantified MPP3 data and Figure S3 shows our gating schemes.

Additionally, we isolated BMDMs from mice fed SC, WD, and KD for 2 weeks, expand and differentiate these cells 7 days and then stimulated with LPS to analyze cytokine release. We now have found BMDMs from KD-fed mice treated ex vivo with LPS after 7 days of macrophage differentiation shows enhanced cytokine production compared to SC- and WD-fed mice (Figure 2D-E), highlighting the intriguing biology behind KD-dependent trained immunity. We have since concluded KD is inducing a trained immunity response and have added this to the title and within the analysis.

References for revision:

1. M. Perreault *et al.*, A distinct fatty acid profile underlies the reduced inflammatory state of metabolically healthy obese individuals. *PLoS One* 9, e88539 (2014).

2. S. A. Abdelmagid *et al.*, Comprehensive Profiling of Plasma Fatty Acid Concentrations in Young Healthy Canadian Adults. *PLoS One*, (2015).

3. D. D. Black, Development and Physiological Regulation of Intestinal Lipid Absorption. I. Development of intestinal lipid absorption: cellular events in chylomicron assembly and secretion. *Am J Physiol Gastrointest Liver Physiol* 293, (2007).

4. C. M. Mansbach II, F. Gorelick, Development and physiological regulation of intestinal lipid absorption. II. Dietary lipid absorption, complex lipid synthesis, and the intracellular packaging and secretion of chylomicrons. *Am J Physiol Gastrointest Liver Physiol* 293, (2007).

5. M. H. Karavolos *et al.*, Adrenaline modulates the global transcriptional profile of *Salmonella* revealing a role in the antimicrobial peptide and oxidative stress resistance responses. *BMC Genomics* 9, 458 (2008).

6. B. A. Napier *et al.*, Complement pathway amplifies caspase-11-dependent cell death and endotoxininduced sepsis severity. *J Exp Med*, (2016).

7. B. A. Napier *et al.*, Western diet regulates immune status and the response to LPS-driven sepsis independent of diet-associated microbiome. *Proc Natl Acad Sci U S A* 116, 3688-3694 (2019).

8. F. B. SCHWEINBURG, H. A. FRANK, J. FINE, Bacterial factor in experimental hemorrhagic shock; evidence for development of a bacterial factor which accounts for irreversibility to transfusion and for the loss of the normal capacity to destroy bacteria. *Am J Physiol* 179, 532-540 (1954).

9. S. JACOB *et al.*, Bacterial action in development of irreversibility to transfusion in hemorrhagic shock in the dog. *Am J Physiol* 179, 523-531 (1954).

10. C. T. Fagundes, D. G. Souza, J. R. Nicoli, M. M. Teixeira, Control of host inflammatory responsiveness by indigenous microbiota reveals an adaptive component of the innate immune system. *Microbes Infect* 13, 1121-1132 (2011).

11. C. T. Fagundes, F. A. Amaral, A. L. Teixeira, D. G. Souza, M. M. Teixeira, Adapting to environmental stresses: the role of the microbiota in controlling innate immunity and behavioral responses. *Immunol Rev* 245, 250-264 (2012).

12. D. G. Souza *et al.*, The essential role of the intestinal microbiota in facilitating acute inflammatory responses. *J Immunol* 173, 4137-4146 (2004).

13. S. L. Foster, D. C. Hargreaves, R. Medzhitov, Gene-specific control of inflammation by TLR-induced chromatin modifications. *Nature* 447, 972-978 (2007).

14. M. Divangahi *et al.*, Trained immunity, tolerance, priming and differentiation: distinct immunological processes. *Nat Immunol* 22, 2-6 (2021).

15. Y. V. Radzyukevich, N. I. Kosyakova, I. R. Prokhorenko, Participation of Monocyte Subpopulations in Progression of Experimental Endotoxemia (EE) and Systemic Inflammation. *Journal of Immunology Research* 2021, 1-9 (2021).

[Editors’ note: what follows is the authors’ response to the second round of review.]

The manuscript has been improved but there are some remaining issues that need to be addressed, as outlined below:The authors have wonderfully addressed the comments, and have resubmitted a very strong and thorough study on the role of dietary fatty acids on the induction of in vivo trained immunity. The manuscript is nearly ready for publication, especially considering the novelty and strength of their findings, not to mention the relevance to the field of trained immunity as well as metainflammation. There are however a few final comments on the present manuscript.The first thing is the so-called 'obesity paradox'. People overweight or with obesity are generally protected from deadly/lethal septicemia when compared to normal or underweight individuals. Two weeks of high-fat feeding may not be sufficient to induce obesity, however, it would be interesting to know if the mice have gained weight. It would be of interest to briefly discuss the clinical conundrum of the 'obesity paradox', especially given the highly lethal phenotype of WD and KD-fed mice in for example Figure 1B.

The reviewer brings up an important point. We have previously addressed this paradox question in our model of endotoxemia and diet-induced weight gain in Napier, *et al.* 2019, PNAS. However, it’s important for the reader to understand the context and application of our studies, thus we have added a paragraph to the Discussion that explores this idea in the context of our findings (Lines 452-463).

With sepsis, the general course of the disease is indeed hyper-inflammation leading to organ failure and death, or immune paralysis and increased risk of mortality via secondary infection. Given that your data suggest that the mice indeed enter into the immune-paralyzed state, do you have any data to show whether the mice died from organ failure or secondary infections? Regardless of the phenotype, WD or KD-fed mice are more susceptible to lethal responses to LPS. Germ-free mice similarly show lethality making the effect clearer to being the result of organ failure/damage. Though, the cytokines, especially IL-10 make this distinction even murkier. The germ-free clearly look as though they are dying from hyper-inflammation, while the BALB/c mice are seemingly dying in the immune-paralyzed state; given they aren't germ-free and are susceptible to secondary infection with commensals. How would the authors rationalize this?

We agree with the reviewer, it is important to discuss the implications of immunoparalysis driving mortality in the endotoxemia model in the context of our findings. Previous seminal studies concluded that mice treated with antibodies to the TNF receptor and challenged with systemic LPS increased survival from 0% to nearly 100%, suggesting that acute inflammation driven by TNF is responsible for endotoxemia-related survival (*1, 2*). Further, it has been shown that TNF is required for acute renal failure (*3*), lung injury (*4*), and liver damage (*5*) during LPS challenge. In agreeance with conventional mouse data, it has also been shown that acute inflammation, specifically TNF production, is a driver of endotoxemia in germ-free (GF) mice (*6*). These data suggest that acute inflammation, specifically the bioactivity of TNF, drives endotoxemia mortality and organ damage in conventional and GF mice. Thus, although our conventional mice show signs of immunoparalysis, we believe that the acute (515h) enhanced *Tnf* in the blood of both conventional and GF mice is responsible for endotoxemia mortality and subsequent organ failure, as seen in previous publications. Considering this, we agree that this is important to mention in the context of our results, thus we have added a section in the Discussion that mentions these previously published data (Lines 501-510).

Sepsis is a medical term that describes an infection of the blood. In your model, LPS is used to induce systemic inflammation. Although many of the symptoms and effects overlap profoundly with those seen in sepsis, we advise against using the term sepsis to describe your model since no infection is established. Endotoxin challenge, or endotoxin stimulation, is more appropriate/accurate semantics.

We agree with the reviewer and though we define our disease model as “LPS-induced acute septic shock”, it is important that the readers understand that we are using the traditional “endotoxemia” model. We have changed all references to this disease model to “endotoxemia”.

Page 17 lines 461 – 463 reading: "Our findings align with the growing body of evidence indicating that trained immunity is a double-edged sword, where the phenomenon can be beneficial for resistance to infection, but detrimental in the context of inflammatory disease." This reads a bit off given that, in fact, your model uses a bacterial-derived stimulus to induce a lethal inflammatory response in WD and KD-'trained' mice. You could hardly call this a beneficial effect of trained immunity.

We agree with the reviewer that this statement as-is is confusing, thus we have changed the wording: “Our findings align with the growing body of evidence indicating that trained immunity is a double-edged sword, where the phenomenon can be beneficial for resistance to infection, but detrimental in the context of diseases exacerbated by systemic inflammation” (Lines 71-73; 472-476), and include a detailed explanation of our statement in the Discussion (Lines 472-476).

A technical hurdle for figure 3 is that the mice are on a ketogenic diet, cells are harvested, and then brought into culture with glucose-enriched media. We wonder then how the sudden overabundance of glucose might contribute to the elevated cytokine responsiveness seen in the data presented in these figure panels, as opposed to the ketogenic diet per se giving rise to increased immune responsiveness. It is worth saying that the in-vivo cytokine responses are much milder than what's seen ex vivo.

The reviewer brings up an interesting and important point. It has been shown that TNF, IL-6, or IL-1b secretion is enhanced in WT BMDMs treated for a short period (48 h) with high-glucose media (*7*). However, high-glucose media does not alter TNF, IL-6, or IL-1b secretion, or mitochondrial activity, in WT BMDMs treated with LPS following 7 d of differentiation in high glucose media (*7*). Thus, since metabolic adaptation takes place within a few days for BMDMs cultured in high glucose media and does not alter inflammatory cytokine secretion following LPS challenge after 7d in high glucose media, we suggest it is unlikely that high glucose media after 7 d contributed to the significant augmentation of LPS-induced TNF and IL-6 secretion for BMDMs from KD-fed mice compared to controls (*7*). Further, the speed of metabolic shifts (or metabolic flexibility) from glycolysis to ketosis (and reversed) are often similar in lean adults (regardless of fat-intake), suggesting that both SC- and KD-fed mice will have similar metabolic flexibility in response to 7 d of high-glucose (*8, 9*). We have not analyzed the exact metabolic flexibility of the mitochondria within SC- or KD-derived BMDMs, but this is an intense area of interest for our lab and we are currently researching this topic.

Lastly, while LPS-challenged macrophages preferentially utilize glucose as their main energy source in order to upregulate expression and release of pro-inflammatory cytokines, there are several variables that may contribute to the mismatch between cytokine mRNA expression in the blood and expression or secretion of cytokines by differentiated immune cells in response to LPS (*10*). Specifically, monocytes (the major producer of systemic cytokines during LPS challenge) are only approximately 4% of the leukocytes in the blood in mice. Thus, we are seeing a diluted response, whereas in vitro we would be directly assessing monocyte (only) responses.

We agree this is an important point to mention within the manuscript and have since added a section in the Discussion covering this topic (Lines 441-451).

References

1. B. Beutler, I. W. Milsark, A. C. Cerami, Passive immunization against cachectin/tumor necrosis factor protects mice from lethal effect of endotoxin. *Science* 229, 869-871 (1985).

2. K. M. Mohler *et al.*, Soluble tumor necrosis factor (TNF) receptors are effective therapeutic agents in lethal endotoxemia and function simultaneously as both TNF carriers and TNF antagonists. *J Immunol* 151, 1548-1561 (1993).

3. P. N. Cunningham *et al.*, Acute renal failure in endotoxemia is caused by TNF acting directly on TNF receptor-1 in kidney. *J Immunol* 168, 5817-5823 (2002).

4. Y. Chen *et al.*, Aerosol synthesis of cargo-filled graphene nanosacks. *Nano Lett* 12, 1996-2002 (2012).

5. W. Zhong *et al.*, Curcumin alleviates lipopolysaccharide induced sepsis and liver failure by suppression of oxidative stress-related inflammation via PI3K/AKT and NF-κB related signaling. *Biomed Pharmacother* 83, 302-313 (2016).

6. D. G. Souza *et al.*, The essential role of the intestinal microbiota in facilitating acute inflammatory responses. *J Immunol* 173, 4137-4146 (2004).

7. T. S. Ayala *et al.*, High glucose environments interfere with bone marrow-derived macrophage inflammatory mediator release, the TLR4 pathway and glucose metabolism. *Scientific Reports*, (2019).

8. D. E. Kelley, B. Goodpaster, R. R. Wing, J. A. Simoneau, Skeletal muscle fatty acid metabolism in association with insulin resistance, obesity, and weight loss. *Am J Physiol* 277, E1130-1141 (1999).

9. R. L. Smith, M. R. Soeters, R. C. I. Wüst, R. H. Houtkooper, Metabolic Flexibility as an Adaptation to Energy Resources and Requirements in Health and Disease. *Endocr Rev* 39, 489-517 (2018).

10. A. J. Freemerman, L. Makowski, Metabolic reprogramming of macrophages

Glucose transporter 1 (GLUT1)-mediated glucose metabolism drives a proinflammatory phenotype. *The Journal of Biological Chemistry* 289, 7884-7896 (2014).